# G-Transformer for Conditional Average Potential Outcome Estimation over Time

## Abstract

Estimating potential outcomes for treatments over time based on observational data is important for personalized decision-making in medicine. Yet, existing neural methods for this task either (1) do not perform proper adjustments for time-varying confounders, or (2) have a problematic adjustment strategy. In order to address both limitations, we introduce the *G-transformer* (GT). Our GT is a novel, neural end-to-end model which adjusts for time-varying confounders in order to estimate conditional average potential outcomes (CAPOs) over time. Specifically, our GT is the first neural model to perform fully regression-based iterative G-computation for CAPOs in the time-varying setting. We evaluate the effectiveness of our GT across various experiments. In sum, this work represents a significant step towards personalized decision-making from electronic health records.

## 1 Introduction

Causal machine learning has recently garnered significant attention with the aim to personalize treatment decisions in medicine (Feuerriegel et al., 2024). Here, an important task is to estimate conditional average potential outcomes (CAPOs) from observational data over time (see Fig. 1). Recently, such data has become prominent in medicine due to the growing prevalence of electronic health records (EHRs) (Allam et al., 2021; Bica et al., 2021) and wearable devices (Battalio et al., 2021; Murray et al., 2016).

Several neural methods have been developed for estimating CAPOs over time. However, existing methods suffer from one of two possible **limitations** (see Table 1): ① Methods **without proper adjustments for time-varying confounding** (Bica et al., 2020; Melnychuk et al., 2022; Seedat et al., 2022) exhibit significant bias, as they do not target the correct estimand. Hence, these methods have irreducible estimation errors irrespective of the amount of available data, which renders them unsuitable for medical applications. ② Existing methods that perform proper time-varying adjustments (Li et al., 2021; Lim et al., 2018) have a **problematic adjustment strategy**. Here, the causal adjustments are based on the estimation of either the distributions of all time-varying covariates, or on inverse propensity weighting at several time steps in the future. While the former is impracticable when granular patient information is available, the latter suffers from strong overlap violations in the time-varying setting. To the best of our knowledge, there is no method that can address both ① and ②.

To fill the above research gap, we propose the *G-transformer* (GT), a novel, neural end-to-end transformer that overcomes both limitations of existing methods. Our GT builds upon G-computation (Bang & Robins, 2005; Robins & Hernán, 2009). However, unlike existing neural models that perform G-computation (Li et al., 2021), our GT is based on an iterative *regression* scheme and does *not* require estimating any probability distribution. As a result, our GT has two clear strengths: it performs ① **proper adjustments for time-varying confounding**, and it is ② **fully regression-based with low-variance pseudo-outcomes**.

Our contributions are three-fold:[1] (1) We introduce the first neural end-to-end method for estimating CAPOs over time with ① **proper adjustments for time-varying confounding**, while ② avoiding

---

[1] Code and data are anonymized in `https://anonymous.4open.science/r/G_transformer-130D`. Upon acceptance, it will be moved to a public Github repository.

| | CRN (Bica et al., 2020) | TE-CDE (Seedat et al., 2022) | CT (Melnychuk et al., 2022) | RMSNs (Lim et al., 2018) | G-Net (Li et al., 2021) | GT (ours) |
|---|---|---|---|---|---|---|
| ① Proper adjustments for time-varying confounding | ✗ | ✗ | ✗ | ✓ | ✓ | ✓ |
| ② Fully regression-based | ✓ | ✓ | ✓ | ✓ | ✗ | ✓ |
| Low-variance pseudo-outcomes | — | — | — | ✗ | — | ✓ |

Table 1: Overview of key neural methods for estimating CAPOs over time. Methods that perform **proper adjustments** for time-varying confounding target the correct causal estimand and, therefore, have no infinite-data bias. Fully regression-based methods avoid estimating high-dimensional probability distributions. Further, we show that IPW generates pseudo-outcomes with larger variance than G-computation (Prop. 3).

a problematic adjustment strategy. (2) To the best of our knowledge, we are the first to leverage regression-based iterative G-computation for estimating CAPOs over time in a neural end-to-end training algorithm. (3) We demonstrate the effectiveness of our GT across various experiments. In the future, we expect our GT to help personalize decision-making from patient trajectories in medicine.

## 2 RELATED WORK

We discuss methods for estimating CAPOs in the static setting, survival analysis with pseudo-outcomes, Q-learning, and other literature streams in an extended related work in Supplement A.

**Estimating APOs over time:** Estimating average potential outcomes (APOs) over time has a long-ranging history in classical statistics and epidemiology (Lok, 2008; Robins, 1986; Rytgaard et al., 2022; van der Laan & Gruber, 2012). Popular approaches are the G-methods (Robins & Hernán, 2009), which include marginal structural models (MSMs) (Robins & Hernán, 2009;

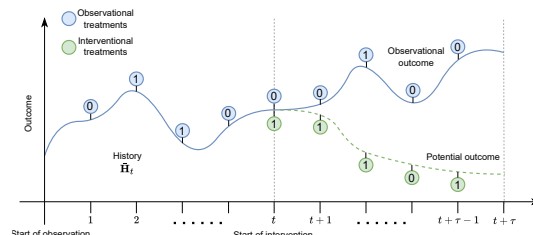

Figure 1: Trajectories with outcomes under *observational* vs. *interventional* treatment sequences.

Robins et al., 2000), structural nested models (Robins, 1994; Robins & Hernán, 2009) and the G-computation (Bang & Robins, 2005; Robins, 1999; Robins & Hernán, 2009), and TMLE (**?**), which involves a targeting step for the APO. G-computation has also been incorporated into neural models such as LSTMs (Frauen et al., 2023a), and TMLE to transformers (Shirakawa et al., 2024). However, all of these works do **not** focus estimating CAPOs. In particular, (Shirakawa et al., 2024) is explicitly **biased** by sequentially targeting the APO and, thereby, ignores individual patient characteristics. Further, it require estimation of additional nuisance such as the propensity score. Finally, it is only evaluated for estimating APOs. As this entire literature stream does not account for individual-level patient characteristics, it serves a different purpose and is thus **not** suitable for personalized decision-making in medicine.

**Estimating CAPOs over time:** In this work, we focus on the task of estimating the heterogeneous response to a sequence of treatments through *conditional average potential outcomes* (CAPOs).[2] Hence, we now summarize key neural methods that have been developed for estimating CAPOs over time (see Table 1). However, these methods fall into two groups with ***important limitations***, as discussed in the following:

Limitation ① proper adjustments: A number of neural methods for estimating CAPOs have been proposed that **do not properly adjust** for time-varying confounders (Bica et al., 2020; Melnychuk et al., 2022; Seedat et al., 2022). Therefore, they are *biased* as they do not target the correct estimand. Here, key examples are the counterfactual recurrent network (CRN) (Bica et al., 2020), the treatment effect neural controlled differential equation (TE-CDE) (Seedat et al., 2022) and the causal transformer (CT) (Melnychuk et al., 2022). These methods try to account for time-varying confounders through balanced representations. However, balancing was originally designed for reducing finite-sample

---

[2]This is frequently known as *counterfactual prediction*. However, our work follows the potential outcomes framework (Neyman, 1923; Rubin, 1978), and we thus use the terminology of CAPO estimation.

estimation variance and *not* for mitigating confounding bias (Shalit et al., 2017). Hence, this is a heuristic and may even introduce another source of representation-induced confounding bias (Melnychuk et al., 2024). Unlike these methods, our GT performs **proper adjustments for time-varying confounders**.

Limitation ② adjustment strategy: Existing neural methods with proper causal adjustments require estimating full probability distributions at several time steps in the future, or inverse propensity weighting, both of which are **problematic adjustment strategies**. Prominent examples are the recurrent marginal structural networks (RMSNs) (Lim et al., 2018) and the G-Net (Li et al., 2021). Here, the RMSNs leverage MSMs (Robins & Hernán, 2009; Robins et al., 2000) and construct pseudo outcomes through inverse propensity weighting (IPW).However, IPW constructs pseudo-outcomes with large variance compared to G-computation as we show in Prop. 3. Further, the G-Net (Li et al., 2021) uses G-computation (Robins, 1999; Robins & Hernán, 2009) to adjust for confounding (see Supplement C). For this, G-Net proceeds by estimating the *entire distribution of all confounders at several time-steps in the future*. Therefore, may suffer from large estimation variance. Different to G-Net, our GT makes use of regression-based G-computation. We discuss key differences in Section 4.4 and Supplement F.

**Research gap:** None of the above neural methods leverages G-computation (Bang & Robins, 2005; Robins, 1999) for estimating CAPOs through iterative regressions. Therefore, to the best of our knowledge, we propose the first neural end-to-end model that ① **properly adjusts for time-varying confounders** through regression-based iterative G-computation. Hence, our GT yields estimates of CAPOs over time that have ② are **fully regression-based** with **low-variance pseudo-outcomes**.

## 3 PROBLEM FORMULATION

**Setup:** We follow previous literature (Bica et al., 2020; Li et al., 2021; Lim et al., 2018; Melnychuk et al., 2022) and consider data that consist of realizations of the following random variables: (i) outcomes $Y_t \in \mathbb{R}^{d_y}$, (ii) covariates $X_t \in \mathbb{R}^{d_x}$, and (iii) treatments $A_t \in \{0,1\}^{d_a}$ at time steps $t \in \{0, \ldots, T\} \subset \mathbb{N}_0$, where $T$ is the time window that follows some unknown counting process. We are interested in estimating CAPOs for $\tau$ steps in the future. For any random variable $U_t \in \{Y_t, X_t, A_t\}$, we write $U_{t:t+\tau} = (U_t, \ldots, U_{t+\tau})$ to refer to a specific subsequence of a random variable. We further write $\bar{U}_t = U_{0:t}$ to denote the full trajectory of $U$ including time $t$. Finally, we write $\bar{H}_{t+\delta}^t = (\bar{Y}_{t+\delta}, \bar{X}_{t+\delta}, \bar{A}_{t-1})$ for $\delta \geq 0$, and we let $\bar{H}_t = \bar{H}_t^t$ denote the collective history of (i)–(iii).

**Estimation task:** We are interested in estimating the *conditional* average potential outcome (CAPO) for a future, interventional sequence of treatments, given the observed history. For this, we build upon the potential outcomes framework (Neyman, 1923; Rubin, 1978) for the time-varying setting (Robins & Hernán, 2009; Robins et al., 2000). Hence, we aim to estimate the potential outcome $Y_{t+\tau}[a_{t:t+\tau-1}]$ at future time $t+\tau$, $\tau \in \mathbb{N}$, for an interventional sequence of treatments $a = a_{t:t+\tau-1}$, *conditionally* on the observed history $\bar{H}_t = \bar{h}_t$. That is, our objective is to estimate

$$\mathbb{E}\left[Y_{t+\tau}[a_{t:t+\tau-1}] \mid \bar{H}_t = \bar{h}_t\right]. \tag{1}$$

**Identifiability:** In order to estimate the causal quantity in Eq. (1) from observational data, we make the following identifiability assumptions (Robins & Hernán, 2009; Robins et al., 2000) that are standard in the literature (Bica et al., 2020; Li et al., 2021; Lim et al., 2018; Melnychuk et al., 2022; Seedat et al., 2022): (1) *Consistency:* For an observed sequence of treatments $\bar{A}_t = \bar{a}_t$, the observed outcome $Y_{t+1}$ equals the corresponding potential outcome $Y_{t+1}[\bar{a}_t]$. (2) *Positivity:* For any history $\bar{H}_t = \bar{h}_t$ that has non-zero probability $\mathbb{P}(\bar{H}_t = \bar{h}_t) > 0$, there is a positive probability $\mathbb{P}(A_t = a_t \mid \bar{H}_t = \bar{h}_t) > 0$ of receiving any treatment $A_t = a_t$, where $a_t \in \{0,1\}^{d_a}$. (3) *Sequential ignorability:* Given a history $\bar{H}_t = \bar{h}_t$, the treatment $A_t$ is independent of the potential outcome $Y_{t+\delta}[a_{t:t+\delta-1}]$, that is, $A_t \perp Y_{t+\delta}[a_{t:t+\delta-1}] \mid \bar{H}_t = \bar{h}_t$ for all $a_{t:t+\delta-1} \in \{0,1\}^{\delta \times d_a}$.

The above assumptions are standard in the literature (Bica et al., 2020; Li et al., 2021; Lim et al., 2018; Melnychuk et al., 2022; Seedat et al., 2022). In clinical scenarios, (i) consistency is typically guaranteed as long as data is properly recorded. Positivity can be guaranteed by filtering the data or by using propensity score clipping. Further, with growing amounts of observational data, this becomes less of a restriction. Finally, relaxations of (iii) ignorability are typically studies in sensitivity

analysis (Frauen et al., 2023b; Oprescu et al., 2023) and partial identification (Duarte et al., 2023), which is orthogonal to our work. We provide a discussion on the applicability in medical scenarios in Supplement B.

**G-computation:** Estimating CAPOs without confounding bias poses a non-trivial challenge in the time-varying setting. The issue lies in the complexity of handling future time-varying confounders. In particular, for $\tau \geq 2$ and $1 \leq \delta \leq \delta' \leq \tau - 1$, future covariates $X_{t+\delta}$ and outcomes $Y_{t+\delta}$ may affect the probability of receiving certain treatments $A_{t+\delta'}$. Importantly, the time-varying confounders are *unobserved* during inference time, which is generally known as *runtime confounding* (Coston et al., 2020). Therefore, in order to estimate the direct effect of an interventional treatment sequence, one needs to adjust for the time-varying confounders. That is, it is in general **insufficient** to only adjust for the history (Frauen et al., 2024) via

$$\mathbb{E}\left[Y_{t+\tau}[a_{t:t+\tau-1}] \mid \bar{H}_t = \bar{h}_t\right] \neq \mathbb{E}\left[Y_{t+\tau} \mid \bar{H}_t = \bar{h}_t, A_{t:t+\tau-1} = a_{t:t+\tau-1}\right]. \tag{2}$$

As a side note, the problem of *time-varying confounding does not arise for one-step ahead predictions* (i.e., $\tau = 1$). Here, under assumptions (i)–(iii), conditioning on the observed history is equivalent to backdoor-adjustments in the static setting.

One way to adjust for time-varying confounders is IPW (Robins & Hernán, 2009; Robins et al., 2000), which is leveraged by RMSNs (Lim et al., 2018). However, as we show in Supplement E, IPW is subject to large variance. Instead, we leverage G-computation (Bang & Robins, 2005; Robins, 1999; Robins & Hernán, 2009), which provides a rigorous way to account for the time-varying confounders through **proper adjustments**. Formally, G-computation identifies the causal quantity in Eq. (1) via

$$\mathbb{E}[Y_{t+\tau}[a_{t:t+\tau-1}] \mid \bar{H}_t = \bar{h}_t]$$
$$= \mathbb{E}\bigg\{\mathbb{E}\bigg[\ldots \mathbb{E}\big\{\mathbb{E}[Y_{t+\tau} \mid \bar{H}^t_{t+\tau-1}, A_{t:t+\tau-1} = a_{t:t+\tau-1}] \mid \bar{H}^t_{t+\tau-2}, A_{t:t+\tau-2} = a_{t:t+\tau-2}\big\} \tag{3}$$
$$\ldots \big| \bar{H}^t_{t+1}, A_{t:t+1} = a_{t:t+1}\bigg] \big| \bar{H}_t = \bar{h}_t, A_t = a_t\bigg\}.$$

A derivation of the G-computation formula for CAPOs is given in Supplement C. However, due to the nested structure of G-computation, estimating Eq. (3) from data is challenging.

So far, only G-Net (Li et al., 2021) has used G-computation for estimating CAPOs in a neural model. For this, G-Net makes a Monte Carlo approximation of Eq. (3) through

$$\int_{\mathbb{R}^{d_x \times \tau-1} \times \mathbb{R}^{d_y \times \tau-1}} \mathbb{E}[Y_{t+\tau} \mid \bar{H}^t_{t+\tau-1} = \bar{h}^t_{t+\tau-1}, A_{t:t+\tau-1} = a_{t:t+\tau-1}]$$
$$\times \prod_{\delta=1}^{\tau-1} p(x_{t+\delta}, y_{t+\delta} \mid \bar{h}_t, x_{t+1:t+\delta-1}, y_{t+1:t+\delta-1}, a_{t:t+\delta-1})\, \mathrm{d}(x_{t+1:t+\tau-1}, y_{t+1:t+\tau-1}). \tag{4}$$

However, Eq. (76) requires estimating the *entire distribution* of all time-varying confounders at several time steps in the future, which may lead to **large estimation variance**. In particular, *all moments* of a $(\tau-1) \times (d_x + d_y)$-dimensional random variable need to be estimated, which leads to estimation of nuisance. We provide more details in Supplement F.

In contrast, our GT does **not** rely on high-dimensional integral approximation through Monte Carlo sampling. Further, our GT does **not** require estimating any probability distribution. Instead, it performs *regression-based iterative G-computation* in an end-to-end transformer architecture. Thereby, we perform **proper adjustments for time-varying confounding** through Eq. (3), while relying only on regressions of via **low-variance pseudo-outcomes**.

## 4 G-TRANSFORMER

In the following, we present our G-transformer. Inspired by (Bang & Robins, 2005; Robins, 1999; Robins & Hernán, 2009) for APOs, we reframe G-computation for CAPOs over time through recursive conditional expectations. Thereby, we precisely formulate the training objective of our GT through iterative regressions. Importantly, existing approaches for estimating APOs do not estimate potential outcomes on an individual level for a given history $\bar{H}_t = \bar{h}_t$, because of which they are **not**

sufficient for estimating CAPOs. Therefore, we proceed below by first extending regression-based iterative G-computation to account for the heterogeneous response to a treatment intervention. We then detail the architecture of our GT and provide details on the end-to-end training and inference.

## 4.1 REGRESSION-BASED ITERATIVE G-COMPUTATION FOR CAPOS

Our GT leverages G-computation as in Eq. (3) and, therefore, properly adjusts for time-varying confounders in Eq. (1). However, we do not attempt to integrate over the estimated distribution of all time-varying confounders. Instead, one of our main novelties is that our GT performs iterative regressions in a *neural end-to-end architecture*. This allows us to estimate Eq. (1) **without estimating high-dimensional probability distributions.**

We reframe Eq. (3) equivalently as a recursion of conditional expectations. Thereby, we can formulate the iterative regression objective of our GT. In particular, our approach resembles an *iterative pseudo-outcome regression*. For this, let

$$g_{t+\delta}^a(\bar{h}_{t+\delta}^t) = \mathbb{E}[G_{t+\delta+1}^a \mid \bar{H}_{t+\delta}^t = \bar{h}_{t+\delta}^t, A_{t:t+\delta} = a_{t:t+\delta}], \tag{5}$$

where the *pseudo-outcomes* are defined as

$$G_{t+\tau}^a = Y_{t+\tau} \tag{6}$$

and

$$G_{t+\delta}^a = g_{t+\delta}^a(\bar{H}_{t+\delta}^t) \tag{7}$$

for $\delta = 0, \ldots, \tau - 1$. By reformulating the G-computation formula through recursions, the nested expectations in Eq. (3) are now given by

$$G_{t+\tau-1}^a = \mathbb{E}[Y_\tau \mid \bar{H}_{t-1}^t, A_{t:t+\tau-1} = a_{t:t+\tau-1}], \tag{8}$$

$$G_{t+\tau-2}^a = \mathbb{E}\Big[\mathbb{E}[Y_\tau \mid \bar{H}_{t-1}^t, A_{t:t+\tau-1} = a_{t:t+\tau-1}] \mid \bar{H}_{t-2}^t, A_{t:t+\tau-2} = a_{t:t+\tau-2}\Big], \tag{9}$$

$$\ldots \tag{10}$$

Hence, the G-computation formula in Eq. (3) can be rewritten as

$$g_t^a(\bar{h}_t) = \mathbb{E}[Y_{t+\tau}[a_{t:t+\tau-1}] \mid \bar{H}_t = \bar{h}_t]. \tag{11}$$

We show in the following proposition that iterative pseudo-outcome regression recovers the CAPOs and thus performs proper adjustments for time-varying confounding. We summarize the iterative pseudo-outcome regression for CAPOs in the following proposition.

**Proposition 1.** *The regression-based iterative G-computation yields the CAPO in Eq. (1).*

*Proof.* See Supplement D.1. □

To further illustrate our regression-based iterative G-computation, we provide **two examples** in Supplement D.3, where we show step-by-step how our approach adjusts for time-varying confounding.

In order to correctly estimate Eq. (2) for a given history $\bar{H}_t = \bar{h}_t$ and an interventional treatment sequence $a = a_{t:t+\tau-1}$, all subsequent pseudo-outcomes in Eq. (7) are required. However, the ground-truth realizations of the pseudo-outcomes $G_{t+\delta}^a$ are *not available in the data*. Instead, only realizations of $G_{t+\tau}^a = Y_{t+\tau}$ in Eq. (6) are observed during the training. Hence, when training our GT, it alternately generates predictions $\tilde{G}_{t+\delta}^a$ of the pseudo-outcomes for $\delta = 0, \ldots \tau - 1$, which it then uses for learning the estimator of Eq. (5).

Therefore, the training of our GT completes two steps in an iterative scheme: First, it runs a (A) *generation step*, where it generates predictions of the pseudo-outcomes Eq. (7). Then, it runs a (B) *learning step*, where it regresses the predictions $\tilde{G}_{t+\delta}^a$ for Eq. (7) and the observed $G_{t+\tau}^a = Y_{t+\tau}$ in Eq. (6) on the history to update the estimator for Eq. (5). Finally, the updated estimators are used again in the next (A) *generation step*. This procedure resembles an iterative pseudo-outcome regression. Thereby, our GT is designed to simultaneously (A) *generate* predictions and (B) *learn* during the training. Both steps are performed in an end-to-end architecture, ensuring that information is shared across time and data is used efficiently. We detail the architecture as well as training and inference of our GT in the following sections.

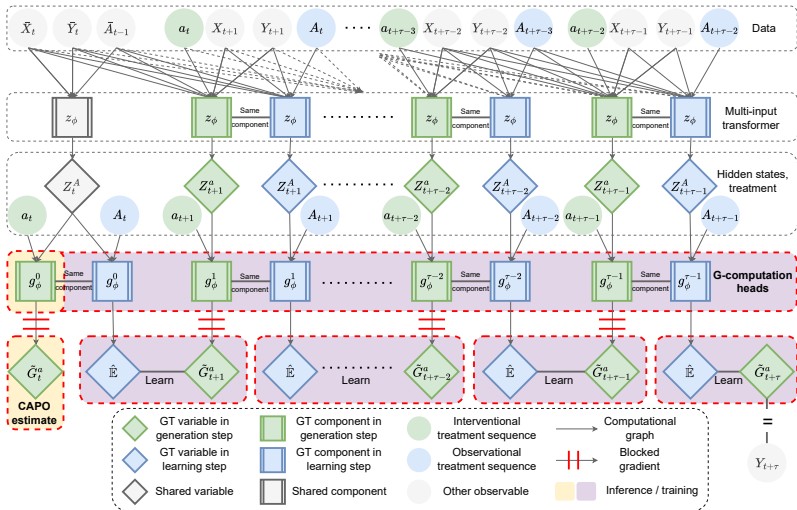

Figure 2: Neural end-to-end architecture and training of our G-transformer.

## 4.2 MODEL ARCHITECTURE

We first introduce the architecture of our GT. Then, we explain the iterative prediction and learning scheme inside our GT, which presents one of the main novelties. Finally, we introduce the inference procedure.

Our GT consists of two key components (see Fig. 2): (i) a *multi-input transformer* $z_\theta(\cdot)$, and (ii) several *G-computation heads* $\{g_\phi^\delta(\cdot)\}_{\delta=0}^{\tau-1}$, where $\theta, \phi$ denote the trainable weights. First, the multi-input transformer encodes the entire observed history. Then, the G-computation heads take the encoded history and perform the iterative regressions according to Eq. (5). We provide further details on the transformer architecture and an illustration in Supplement J. For all $t = 1, \ldots, T - \tau$ and $\delta = 0, \ldots, \tau - 1$, the components are designed as follows:

(i) *Multi-input transformer:* The backbone of our GT is a multi-input transformer $z_\theta(\cdot)$, which consists of three connected encoder-only sub-transformers $z_\theta^k(\cdot)$, $k \in \{1, 2, 3\}$ and is directly inspired by (Melnychuk et al., 2022). We provide details on the architecture in Supplement J. At time $t$, the transformer $z_\theta(\cdot)$ receives data $\bar{H}_t = (\bar{Y}_t, \bar{X}_t, \bar{A}_{t-1})$ as input and passes them to one corresponding sub-transformer. In particular, each sub-transformer $z_\theta^k(\cdot)$ is responsible to focus on one particular $\bar{U}_t^k \in \{\bar{Y}_t, \bar{X}_t, \bar{A}_{t-1}\}$ in order to effectively process the different types of inputs. Further, we ensure that information is shared between the sub-transformers, as we detail below. The output of the multi-input transformer are hidden states $Z_t^A$, which are then passed to the (ii) G-computation heads.

(ii) *G-computation heads:* The *G-computation heads* $\{g_\phi^\delta(\cdot)\}_{\delta=0}^{\tau-1}$ are the read-out component of our GT. As input at time $t + \delta$, the G-computation heads receive the hidden state $Z_{t+\delta}^A$ from the above multi-input-transformer. Recall that we seek to perform the iterative regressions in Eq. (5) and Eq. (2), respectively. For this, we require estimators of $\mathbb{E}[G_{t+\delta+1}^a \mid \bar{H}_{t+\delta}, \bar{A}_{t+\delta}]$. Hence, the G-computation heads compute

$$\hat{\mathbb{E}}[G_{t+\delta+1}^a \mid \bar{H}_{t+\delta}, A_{t+\delta}] = g_\phi^\delta(Z_{t+\delta}^A, A_{t+\delta}), \tag{12}$$

where

$$Z_{t+\delta}^A = z_\theta(\bar{H}_{t+\delta}) \tag{13}$$

for $\delta = 0, \ldots, \tau - 1$. As a result, the G-computation heads and the multi-input transformer together give the estimators that are required for the regression-based iterative G-computation. In particular, we thereby ensure that, for $\delta = 0$, the last G-computation head $g_\phi^0(\cdot)$ is trained as the estimator for the CAPO as given in Eq. (2). That is, for a fully trained multi-input transformer and G-computation heads, our GT estimates the CAPO via

$$\hat{\mathbb{E}}[Y_{t+\tau}[a_{t:t+\tau-1}] \mid \bar{H}_t = \bar{h}_t] = g_\phi^0(z_\theta(\bar{h}_t), a_t). \tag{14}$$

### 4.3 ITERATIVE TRAINING AND INFERENCE TIME

We now introduce the iterative training of our GT, which consists of a (A) *generation step* and a (B) *learning step*. Then, we show how inference for a given history $\bar{H}_t = \bar{h}_t$ can be achieved. We provide pseudocode in Supplement K.

**Iterative training:** Our GT is designed to estimate the CAPO $g_t^a(\bar{h}_t)$ in Eq. (2) for a given history $\bar{H}_t = \bar{h}_t$ and an interventional treatment sequence $a = a_{t:t+\tau-1}$ via Eq. (14). Therefore, the G-computation heads in Eq. (12) require the pseudo-outcomes $\{G_{t+\delta}^a\}_{\delta=1}^{\tau}$ from Eq. (7) during training. However, they are only available in the training data for $\delta = \tau$. That is, we only observe the factual outcomes $G_{t+\tau}^a = Y_\tau$.

As a remedy, our GT first predicts the remaining pseudo-outcomes $\{G_{t+\delta}^a\}_{\delta=1}^{\tau-1}$ in the (A) *generation step*. Then, it can use these generated pseudo-outcomes and the observed $G_{t+\tau}^a$ for learning the network weights $\phi$ in the (B) *learning step*. In the following, we write $\{\tilde{G}_{t+\delta}^a\}_{\delta=1}^{\tau-1}$ for the generated pseudo-outcomes and, for notational convenience, we also write $\tilde{G}_{t+\tau}^a = G_{t+\tau}^a$.

(A) *Generation step:* In this step, our GT generates $\tilde{G}_{t+\delta}^a \approx g_{t+\delta}^a(\bar{H}_{t+\delta}^t)$ as substitutes for Eq. (7), which are the pseudo-outcomes in the iterative regression-based G-computation. Formally, our GT predicts these via

$$\tilde{G}_{t+\delta}^a = g_\phi^\delta(Z_{t+\delta}^a, a_{t+\delta}), \tag{15}$$

where

$$Z_{t+\delta}^a = z_\theta(\bar{H}_{t+\delta}^t, a_{t:t+\delta-1}), \tag{16}$$

for $\delta = 0, \ldots, \tau - 1$. For this, all operations are *detached* from the computational graph. Hence, our GT now has pseudo-outcomes $\{\tilde{G}_{t+\delta}^a\}_{\delta=0}^{\tau}$, which it can use in the following (B) *learning step*. Of note, these generated pseudo-outcomes will be noisy for early training epochs. However, as training progresses, the G-computation heads perform increasingly more accurate predictions, as we explain below.

(B) *Learning step:* This step is responsible for updating the weights $\phi$ of the multi-input transformer $z_\theta(\cdot)$ and the G-computation heads $\{g_\phi^\delta(\cdot)\}_{\delta=0}^{\tau-1}$. For this, our GT learns the estimator for Eq. (5) via

$$\hat{\mathbb{E}}[G_{t+\delta+1}^a \mid \bar{H}_{t+\delta}^t, A_{t:t+\delta}] = g_\phi^\delta(Z_{t+\delta}^A, A_{t+\delta}), \tag{17}$$

where

$$Z_{t+\delta}^A = z_\theta(\bar{H}_{t+\delta}) \tag{18}$$

for $\delta = 0, \ldots, \tau - 1$. In particular, the estimator is optimized by backpropagating the squared error loss $\mathcal{L}$ for all $\delta = 0, \ldots, \tau - 1$ and $t = 1, \ldots, T - \tau$ via

$$\mathcal{L} = \frac{1}{T-\tau} \sum_{t=1}^{T-\tau} \left( \frac{1}{\tau} \sum_{\delta=0}^{\tau-1} \left( g_\phi^\delta(Z_{t+\delta}^A, A_{t+\delta}) - \tilde{G}_{t+\delta+1}^a \right)^2 \right). \tag{19}$$

Then, after $\phi$ is updated, we can use the updated estimator in the next (A) *generation step*.

Here, it is important that for $\delta = \tau$, the pseudo-outcome $\tilde{G}_{t+\tau}^a = Y_{t+\tau}$ is *available in the data*. By estimating $Y_{t+\tau}$ with $g_\phi^{\tau-1}(Z_{t+\tau-1}^A, A_{t+\tau-1})$, it is ensured the last G-computation head $g_\phi^{\tau-1}(\cdot)$ is learned on a ground-truth quantity. Thereby, the weights of $g_\phi^{\tau-1}(\cdot)$ are gradually optimized during training. Hence, the predicted pseudo-outcome

$$\tilde{G}_{t+\tau-1}^a = g_\phi^{\tau-1}(Z_{t+\tau-1}^a, a_{t+\tau-1}) \tag{20}$$

in the next (A) *generation step* become mores accurate. Therefore, the G-computation head $g_\phi^{\tau-2}(\cdot)$ is learned on a more accurate prediction in the following (B) *learning step*, which thus leads to a better generated pseudo-outcome $\tilde{G}_{t+\tau-2}^a$, and so on. As a result, the optimization of the G-computation heads gradually improves from $g_\phi^{\tau-1}(\cdot)$ up to $g_\phi^0(\cdot)$.

**Inference at runtime:** Finally, we introduce how inference is achieved with our GT. Given a history $\bar{H}_t = \bar{h}_t$ and an interventional treatment sequence $a = a_{t:t+\tau-1}$, our GT is trained to estimate of Eq. (1) through Eq. (2). For this, our GT computes the CAPO via

$$\hat{g}_t^a(\bar{h}_t) = \hat{\mathbb{E}}[G_{t+1}^a \mid \bar{H}_t = \bar{h}_t, A_t = a_t] = g_\phi^0(z_\theta(\bar{h}_t), a_t). \tag{21}$$

We summarize this in the following proposition.

**Proposition 2.** *Our GT estimates the G-computation formula as in Eq.* (2) *and, therefore, performs proper adjustments for time-varying confounders.*

*Proof.* See Supplement D.2. □

### 4.4 ADVANTAGES OVER EXISTING APPROACHES

In the following, we explain the differences of our GT compared to (i) CT (Melnychuk et al., 2022) and (ii) G-Net (Li et al., 2021), and (iii) RMSNs (Lim et al., 2018). Importantly, our method has an entirely *different* learning algorithm that allows for *proper adjustments* in an *end-to-end* approach through iterative regressions.

Our GT vastly differs from CT (Melnychuk et al., 2022). Recall that CT does **not** perform proper adjustments for time-varying confounding. In particular, CT targets $\mathbb{E}[Y_{t+\tau} \mid H_t = h_t, A_{t:t+\tau} = a_{t:t+\tau}]$, which is **not** the CAPO (Frauen et al., 2024). Hence, it targets an *incorrect estimand*, leading to irreducible *bias*. Therefore, deploying it to medical scenarios would be irresponsible. In contrast, our GT leverages iterative regression based on the G-computation to *correctly* target the CAPO (Prop. 2). To achieve this, we propose a new generation-learning approach inside our GT.

Our GT is also vastly different from G-Net (Li et al., 2021). In order to estimate a $\tau$-step-ahead CAPO, G-Net requires (i) a $d_y$-dimensional regression as well as estimating the *entire distribution* of a $(\tau - 1) \times (d_y + d_x)$-dimensional confounding variable. That is, it needs to estimate *all moments* of a *high-dimensional* random variable. In contrast, our GT only requires $\tau$ regressions of a $d_y$-dimensional outcome and, hence, only needs to estimate the *first moment* of a much *lower dimensional* random variable. Compared to G-Net, our estimation strategy is unproblematic as it does **not** fit unnecessary nuisance. We provide a detailed comparison in Supplement F.

Finally, RMSNs (Lim et al., 2018) also rely on pseudo-outcome regressions. However, their pseudo-outcomes are constructed via inverse propensity weighting, which leads to pseudo-outcomes with larger variance than ours:

**Proposition 3.** *Pseudo-outcomes constructed via inverse propensity weighting have larger variance than pseudo-outcomes in our G-transformer.*

*Proof.* See Supplement E. □

## 5 EXPERIMENTS

We show the performance of our GT against key neural methods for estimating CAPOs over time (see Table 1). Further details (e.g., implementation details, hyperparameter tuning, runtime) are given in Supplement L. We report ablation studies of our GT in Supplement G.1.

### 5.1 SYNTHETIC DATA

First, we follow common practice in benchmarking for causal inference (Bica et al., 2020; Li et al., 2021; Lim et al., 2018; Melnychuk et al., 2022) and evaluate the performance of our GT against other baselines on fully synthetic data. The use of synthetic data is beneficial as it allows us to simulate the outcomes under a sequence of interventions, which are unknown in real-world datasets. Thereby, we are able to evaluate the performance of all methods for estimating CAPOs over time. Here, our main aim is to show that our GT is *robust against increasing levels of confounding*.

| | $\gamma = 10$ | $\gamma = 11$ | $\gamma = 12$ | $\gamma = 13$ | $\gamma = 14$ | $\gamma = 15$ | $\gamma = 16$ | $\gamma = 17$ | $\gamma = 18$ | $\gamma = 19$ | $\gamma = 20$ |
|---|---|---|---|---|---|---|---|---|---|---|---|
| CRN (Bica et al., 2020) | $4.05 \pm 0.55$ | $5.45 \pm 1.68$ | $6.17 \pm 1.27$ | $4.98 \pm 1.49$ | $5.24 \pm 0.33$ | $4.84 \pm 0.95$ | $5.41 \pm 1.20$ | $5.09 \pm 0.77$ | $5.08 \pm 0.87$ | $4.47 \pm 0.84$ | $4.80 \pm 0.70$ |
| TE-CDE (Seedat et al., 2022) | $4.08 \pm 0.54$ | $4.21 \pm 0.42$ | $4.33 \pm 0.11$ | $4.48 \pm 0.47$ | $4.39 \pm 0.38$ | $4.67 \pm 0.65$ | $4.84 \pm 0.46$ | $4.31 \pm 0.38$ | $4.44 \pm 0.53$ | $4.61 \pm 0.42$ | $4.72 \pm 0.45$ |
| CT (Melnychuk et al., 2022) | $3.44 \pm 0.73$ | $3.70 \pm 0.77$ | $3.60 \pm 0.62$ | $3.87 \pm 0.68$ | $3.88 \pm 0.75$ | $3.87 \pm 0.65$ | $5.26 \pm 1.67$ | $4.04 \pm 0.74$ | $4.13 \pm 0.90$ | $4.30 \pm 0.72$ | $4.49 \pm 0.94$ |
| RMSNs (Lim et al., 2018) | $3.34 \pm 0.20$ | $3.41 \pm 0.17$ | $3.61 \pm 0.25$ | $3.76 \pm 0.25$ | $3.92 \pm 0.26$ | $4.22 \pm 0.40$ | $4.30 \pm 0.52$ | $4.48 \pm 0.59$ | $4.60 \pm 0.46$ | $4.47 \pm 0.53$ | $4.62 \pm 0.51$ |
| G-Net (Li et al., 2021) | $3.51 \pm 0.37$ | $3.71 \pm 0.33$ | $3.80 \pm 0.29$ | $3.89 \pm 0.27$ | $3.91 \pm 0.26$ | $3.94 \pm 0.26$ | $4.05 \pm 0.37$ | $4.09 \pm 0.41$ | $4.22 \pm 0.53$ | $4.21 \pm 0.55$ | $4.24 \pm 0.45$ |
| **GT** (ours) | $3.13 \pm 0.22$ | $3.16 \pm 0.14$ | $3.31 \pm 0.20$ | $3.27 \pm 0.14$ | $3.30 \pm 0.11$ | $3.49 \pm 0.30$ | $3.53 \pm 0.26$ | $3.50 \pm 0.26$ | $3.41 \pm 0.29$ | $3.59 \pm 0.21$ | $3.71 \pm 0.27$ |
| Rel. improvement | 6.4% | 7.3% | 7.9% | 12.9% | 15.0% | 9.9% | 12.9% | 13.1% | 17.4% | 14.8% | 12.5% |

Table 2: RMSE on synthetic data based on the tumor growth model with $\tau = 2$. Our GT consistently outperforms all baselines. We highlight the relative improvement over the best-performing baseline. Reported: average RMSE $\pm$ standard deviation over five seeds.

*Our main evaluation metric is the root mean squared error (RMSE), which is the appropriate evaluation metric for estimating CAPOs and is standard in the literature (Bica et al., 2020; Li et al., 2021; Lim et al., 2018; Melnychuk et al., 2022). Of note, all baselines and our GT are inherently designed for CAPO estimation. Hence, the best-performing method for estimating CAPOs is immediately the best at estimating conditional average treatment effects CATEs).*

**Setting:** For this, we use data based on the pharmacokinetic-pharmacodynamic tumor growth model (Geng et al., 2017), which is a standard dataset for benchmarking causal inference methods in the time-varying setting (Bica et al., 2020; Li et al., 2021; Lim et al., 2018; Melnychuk et al., 2022). Here, the outcome $Y_t$ is the volume of a tumor that evolves according to the stochastic process $Y_{t+1} = \left(1 + \rho \log\left(\frac{K}{Y_t}\right) - \alpha_c c_t - (\alpha_r d_t + \beta_r d_t^2) + \epsilon_t\right) Y_t$, where $\alpha_c$, $\alpha_r$, and $\beta_r$ control the strength of chemo- and radiotherapy, respectively, and where $K$ corresponds to the carrying capacity, and where $\rho$ is the growth parameter. The radiation dosage $d_t$ and chemotherapy drug concentration $c_t$ are applied with probabilities $\sigma(\gamma/D_{\max}(\bar{D}_{15}(\bar{Y}_{t-1} - \bar{D}_{\max}/2))$, where $D_{\max}$ is the maximum tumor volume, $\bar{D}_{15}$ the average tumor diameter of the last 15 time steps, and $\gamma$ controls the confounding strength. We use the same parameterization as in (Melnychuk et al., 2022). For training, validation, and testing, we sample $N = 1000$ trajectories of lengths $T \leq 30$ each.

We are interested in the performance of our GT for increasing levels of confounding. We thus increase the confounding from $\gamma = 10$ to $\gamma = 20$. For each level of confounding, we fix an arbitrary intervention sequence and simulate the outcomes under this intervention for testing.

**Results:** Table 2 shows the average RMSE over five different runs for a prediction horizon of $\tau = 2$. Of note, we emphasize that our comparison is fair (see hyperparameter tuning in Supplement L.1). We make the following observations:

First, our **GT** outperforms all baselines by a significant margin. Importantly, as our GT performs proper adjustments for time-varying confounding, it is robust against increasing $\gamma$. In particular, our GT achieves a performance improvement over the best-performing baseline of up to $17.4\%$. Further, our GT is highly stable, as can be seen by low standard deviation in the estimates, especially compared to the baselines. In sum, our GT performs best in estimating the CAPOs, especially under increasing confounding strength.

Second, the ① baselines that do not perform proper adjustments (i.e., **CRN** (Bica et al., 2020), **TE-CDE** (Seedat et al., 2022), and **CT** (Melnychuk et al., 2022)) exhibit large variations in performance and are thus highly unstable. This is expected, as they do not target the correct causal estimand and, accordingly, suffer from the increasing confounding.

Third, the baselines with ② problematic adjustment strategies (i.e., **RMSNs** (Lim et al., 2018) and **G-Net** (Li et al., 2021)) are slightly more stable than the no-adjustment baselines. This can be attributed to that the tumor growth model has no time-varying covariates $X_t$ and to that we are only focusing on $\tau = 2$-step ahead predictions, both of which reduce the variance. However, the RMSNs and G-Net are still significantly worse than the estimates provided by our GT.

## 5.2 Semi-synthetic data

Next, we study how our GT performs when (i) the covariate space is *high-dimensional* and when (ii) the *prediction windows $\tau$ become larger*. For this, we use semi-synthetic data, which, similar to the fully-synthetic dataset allows us to access the ground-truth outcomes under an interventional sequence of treatments for benchmarking.

| | $\tau=2$ | $\tau=3$ | $\tau=4$ | $\tau=5$ | $\tau=6$ | $\tau=2$ | $\tau=3$ | $\tau=4$ | $\tau=5$ | $\tau=6$ | $\tau=2$ | $\tau=3$ | $\tau=4$ | $\tau=5$ | $\tau=6$ |
|---|---|---|---|---|---|---|---|---|---|---|---|---|---|---|---|
| | | | $N=1000$ | | | | | $N=2000$ | | | | | $N=3000$ | | |
| CRN (Bica et al., 2020) | $0.42\pm0.11$ | $0.58\pm0.21$ | $0.74\pm0.31$ | $0.84\pm0.42$ | $0.95\pm0.51$ | $0.39\pm0.12$ | $0.50\pm0.14$ | $0.58\pm0.15$ | $0.64\pm0.16$ | $0.70\pm0.17$ | $0.37\pm0.10$ | $0.46\pm0.11$ | $0.56\pm0.13$ | $0.65\pm0.16$ | $0.75\pm0.24$ |
| TE-CDE (Seedat et al., 2022) | $0.76\pm0.09$ | $0.91\pm0.15$ | $1.07\pm0.22$ | $1.15\pm0.25$ | $1.24\pm0.28$ | $0.76\pm0.16$ | $0.87\pm0.17$ | $0.98\pm0.17$ | $1.06\pm0.18$ | $1.14\pm0.19$ | $0.71\pm0.09$ | $0.78\pm0.09$ | $0.88\pm0.11$ | $0.94\pm0.12$ | $1.02\pm0.13$ |
| CT (Melnychuk et al., 2022) | $0.33\pm0.14$ | $0.44\pm0.18$ | $0.53\pm0.21$ | $0.57\pm0.19$ | $0.60\pm0.19$ | $0.31\pm0.11$ | $0.41\pm0.13$ | $0.49\pm0.15$ | $0.55\pm0.15$ | $0.60\pm0.15$ | $0.32\pm0.10$ | $0.40\pm0.11$ | $0.49\pm0.12$ | $0.55\pm0.13$ | $0.61\pm0.15$ |
| RMSNs (Lim et al., 2018) | $0.57\pm0.16$ | $0.73\pm0.20$ | $0.87\pm0.22$ | $0.94\pm0.20$ | $1.02\pm0.20$ | $0.62\pm0.25$ | $0.73\pm0.21$ | $0.85\pm0.25$ | $0.96\pm0.26$ | $1.05\pm0.28$ | $0.66\pm0.27$ | $0.76\pm0.24$ | $0.86\pm0.23$ | $0.93\pm0.21$ | $1.00\pm0.20$ |
| G-Net (Li et al., 2021) | $0.56\pm0.14$ | $0.73\pm0.17$ | $0.86\pm0.18$ | $0.95\pm0.20$ | $1.03\pm0.21$ | $0.55\pm0.12$ | $0.73\pm0.14$ | $0.87\pm0.18$ | $1.00\pm0.22$ | $1.12\pm0.26$ | $0.54\pm0.11$ | $0.72\pm0.16$ | $0.88\pm0.21$ | $1.00\pm0.26$ | $1.11\pm0.32$ |
| **GT** (ours) | $0.30\pm0.07$ | $0.36\pm0.11$ | $0.44\pm0.13$ | $0.47\pm0.12$ | $0.54\pm0.13$ | $0.27\pm0.07$ | $0.32\pm0.09$ | $0.38\pm0.10$ | $0.42\pm0.08$ | $0.45\pm0.10$ | $0.24\pm0.07$ | $0.31\pm0.08$ | $0.36\pm0.09$ | $0.42\pm0.10$ | $0.48\pm0.10$ |
| Rel. improvement | 9.5% | 19.7% | 16.3% | 16.7% | 10.8% | 15.3% | 22.5% | 22.5% | 22.6% | 25.0% | 26.7% | 24.0% | 25.2% | 24.6% | 21.6% |

Table 3: RMSE on semi-synthetic data based on the MIMIC-III extract. Our GT consistently outperforms all baselines. We highlight the relative improvement over the best-performing baseline. Reported: average RMSE $\pm$ standard deviation over five seeds.

**Setting:** We build upon the MIMIC-extract (Wang et al., 2020), which is based on the MIMIC-III dataset (Johnson et al., 2016). Here, we use $d_x = 25$ different vital signs as time-varying covariates and as well as gender, ethnicity, and age as static covariates. Then, we simulate observational outcomes for training and validation, and interventional outcomes for testing, respectively. Our data-generating process is taken from (Melnychuk et al., 2022), which we refer to for more details. In summary, the data generation consists of three steps: (1) $d_y = 2$ untreated outcomes $\tilde{Y}_t^j$, $j = 1, 2$, are simulated according to $\tilde{Y}_t^j = \alpha_s^j \text{B-spline}(t) + \alpha_g^j g^j(t) + \alpha_f^j f_Y^j(X_t) + \epsilon_t$, where $\alpha_s^j$, $\alpha_g^j$ and $\alpha_f^j$ are weight parameters, $\text{B-spline}(t)$ is sampled from a mixture of three different cubic splines, and $f_Y^j(\cdot)$ is a random Fourier features approximation of a Gaussian process. (2) A total of $d_a = 3$ synthetic treatments $A_t^l$, $l = 1, 2, 3$, are applied with probability $\sigma(\gamma_Y^l Y_{t-1}^{A,l} + \gamma_X^l f_Y^l(X_t) + b^l)$ where $\gamma_Y^l$ and $\gamma_X^l$ are fixed parameters that control the confounding strength for treatment $A^l$, $Y_t^{A,l}$ is an averaged subset of the previous $l$ treated outcomes, $b^l$ is a bias term, and $f_Y^l(\cdot)$ is a random function that is sampled from an RFF (random Fourier features) approximation of a Gaussian process. (3) The treatments are applied to the untreated outcomes via

$$Y_t^j = \tilde{Y}_t^j + \sum_{i=t-\omega^l}^{t} \frac{\min_{l=1,\ldots,d_a} \mathbb{1}_{\{A_i^l=1\}} p_i^l \beta^{l,j}}{(\omega^l - i)^2}, \tag{22}$$

where $\omega^l$ is the effect window for treatment $A^l$ and $\beta^{l,j}$ controls the maximum effect of treatment $A^l$.

We run different experiments for training, testing, and validation sizes of $N = 1000$, $N = 2000$, and $N = 3000$, respectively, and set the time window to $30 \leq T \leq 50$. As the covariate space is high-dimensional, we thereby study how robust our GT is with respect to estimation variance.

**Results:** Table 3 shows the average RMSE over five different runs. Again, we emphasize that our comparison is fair (see hyperparameter tuning in Supplement L). We make three observations:

First, our **GT** consistently outperforms all baselines by a large margin. The performance of GT is robust across all sample sizes $N$. Further, it is stable across different prediction windows $\tau$. We observe that our GT has a better performance compared to the strongest baseline of up to 26.7%. Further, the results show the clear benefits of our GT in high-dimensional covariate settings and for longer prediction windows $\tau$. In addition, our GT is highly stable, as its estimates exhibit the lowest standard deviation among all baselines. In sum, our GT consistently outperforms all the baselines.

Second, ① baselines that do not perform proper adjustments (i.e., **CRN** (Bica et al., 2020), **CT** (Melnychuk et al., 2022)) tend to perform better than baselines with problematic adjustment strategies (i.e., RMSNs (Lim et al., 2018), G-Net (Li et al., 2021)). The reason is that the former baselines are (i) regression-based (ii) do not require IPW pseudo-outcomes. Hence, they can better handle the high-dimensional covariate space. They are, however, biased as they do not adjust for time-varying confounders and thus still perform significantly worse than our GT.

Third, baselines with ② problematic adjustment strategies (i.e., **RMSNs** (Lim et al., 2018), **G-Net** (Li et al., 2021)) struggle with the high-dimensional covariate space and larger prediction windows $\tau$. This can be expected, as RMSNs suffer from overlap violations and thus produce unstable inverse propensity weights. Similarly, G-Net suffers from the curse of dimensionality, as it requires estimating a $(d_x + d_y) \times (\tau - 1)$-dimensional distribution.

**Conclusion:** In this paper, we propose the GT, a novel end-to-end method that adjusts for time-varying confounding, while avoiding problematic adjustment strategies for estimating of CAPOs. For this, we propose a regression-based learning algorithm that sets our GT apart from existing baselines. Therefore, we expect our GT to be an important step toward personalized medicine.

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

# A EXTENDED RELATED WORK

**Estimating CAPOs in the static setting:** Extensive work on estimating potential outcomes focuses on the *static* setting (e.g., Alaa & van der Schaar, 2017; Frauen et al., 2023b; Johansson et al., 2016; Louizos et al., 2017; Melnychuk et al., 2023; Yoon et al., 2018; Zhang et al., 2020)). However, observational data such as electronic health records (EHRs) in clinical settings are typically measured *over time* (Allam et al., 2021; Bica et al., 2021). Additionally, treatments are rarely applied all at once but rather sequentially over time (Apperloo et al., 2024). Therefore, the underlying assumption of these methods prohibitive and does not properly reflect medical reality. Hence, static methods are **not** tailored to accurately estimate potential outcomes when (i) time series data is observed and (ii) multiple treatments in the future are of interest.

Additional literature on estimating CAPOs over time: There are some non-parametric methods for this task (Schulam & Saria, 2017; Soleimani et al., 2017; Xu et al., 2016), yet these suffer from poor scalability and have limited flexibility regarding the outcome distribution, the dimension of the outcomes, and static covariate data; because of that, we do not explore non-parametric methods further but focus on neural methods instead.[3]

**Survival analysis:** Some works in survival analysis (Andersen & Perme, 2010; Andersen et al., 2017; Su et al., 2022) employ pseudo-outcomes, which is similar to our approach. However, these works are different in that they are aimed at *survival outcomes* and **not** CAPOs for sequences of treatments. Further, they do **not** consider neural networks as estimators. Additionally, (Andersen et al., 2017) only considers a **single, static treatment**, and (Andersen & Perme, 2010) only uses **linear** estimators. Finally, (Su et al., 2022) focuses on **average** causal effects and is therefore not applicable to personalized medicine.

**G-computation and Q-learning:** Q-learning (Murphy, 2003; Kallus & Uehara, 2019) from the reinforcement learning literature (Furuta et al., 2022; Jang et al., 2022; Kumar et al., 2019; Pashevich et al., 2021) is closely related to G-computation, although both have a different purpose. They are similar in that they share a common goal of understanding the effect of treatments/actions, but operate in complementary domains: G-computation is grounded in causal inference for evaluating potential outcomes, whereas Q-learning is rooted in reinforcement learning to derive *policies that maximize long-term rewards*. We show more details on the two in the following:

G-computation can be written as the iterative update

$$g_{t+\delta}^a(\bar{h}_{t+\delta}^t) = \mathbb{E}[G_{t+\delta+1}^a \mid \bar{H}_{t+\delta}^t = \bar{h}_{t+\delta}^t, A_{t:t+\delta} = a_{t:t+\delta}], \tag{23}$$

In our setting, we aim to estimate $\mathbb{E}\left[Y_{t+\tau}[a_{t:t+\tau-1}] \mid \bar{H}_t = \bar{h}_t\right]$.

However, we could also consider the expected *cumulative rewards* $\mathbb{E}\left[\bar{Y}_{t+\tau}[a_{t:t+\tau-1}] \mid \bar{H}_t = \bar{h}_t\right]$, where we define $\bar{Y}_{t+\tau}[a_{t:t+\tau-1}] = \sum_{\ell=1}^{t+\tau} \gamma^\ell Y_{t+\ell}[a_{t:t+\ell-1}]$ and where $\gamma < 1$ is a so-called discount factor that weighs the importance of immediate and future rewards. One can show that the G-computation update becomes

$$g_{t+\delta}^a(\bar{h}_{t+\delta}^t) = \mathbb{E}[Y_{t+\delta} + \gamma G_{t+\delta+1}^a \mid \bar{H}_{t+\delta}^t = \bar{h}_{t+\delta}^t, A_{t:t+\delta} = a_{t:t+\delta}]. \tag{24}$$

If we only care about the *optimal* treatment sequence $a^*$ (i.e., the one that maximizes the cumulative reward), we can write

$$g_{t+\delta}^{a^*}(\bar{h}_{t+\delta}^t) = \mathbb{E}[Y_{t+\delta} + \gamma \max_{a_{t+\delta+1}^*} G_{t+\delta+1}^{a^*} \mid \bar{H}_{t+\delta}^t = \bar{h}_{t+\delta}^t, A_{t:t+\delta} = a_{t:t+\delta}^*]. \tag{25}$$

Eq. (25) is known as *Q-learning* in the literature on dynamic treatment regimes (Murphy, 2003; Kallus & Uehara, 2019) and can be used to compute an optimal dynamic policy.

In reinforcement learning, one often makes *additional* Markov and stationarity assumptions such that the history $\bar{h}_{t+\delta}^t$ simplifies to a single state $s_{t+\delta}$ and the function $g^{a_t^*}(s_t)$ is not dependent on time. These assumptions allow us to consider infinite time-horizons and break the so-called curse of horizon (Kallus & Uehara, 2022; Uehara et al., 2022). Then, Q-learning simplifies to

$$g^{a_t^*}(s_t) = \mathbb{E}[Y_t + \gamma \max_{a_{t+1}^*} G^{a^*} \mid S_t = s_t, A_t = a_t^*], \tag{26}$$

---

[3]Other works are orthogonal to ours. For example, (Hess et al., 2024; Vanderschueren et al., 2023) are approaches for informative sampling and uncertainty quantification, respectively. However, they do not focus on the causal structure in the data, and are therefore *not* primarily designed for our task of interest.

which is often called *fitted Q-iteration* in the RL literature (Kallus & Uehara, 2020; Uehara et al., 2022). In contrast, our work does not make these assumptions.

State-of-the-art neural instantiations such as (Chebotar et al., 2023) are *different* to our work in that they (i) serve the purpose of *learning long-term rewards*, and (ii) rely on *restrictive Markov* assumptions. In contrast, our GT is designed to estimate CAPOs for sequences of treatments, conditionally on the entire individual patient history.

## B  DISCUSSION ON ESTIMATING OUTCOMES FOR SEQUENCES OF TREATMENTS IN MEDICAL SCENARIOS

In this study, we present a novel neural network, the G-transformer, for estimating conditional average potential outcomes (CAPOs) from observational data such as electronic health records (EHRs). Our GT addresses a *crucial question in personalized medicine*: "What would the outcome be for patient X if they were administered treatments A, B, and C sequentially over the next 5 days, given their unique clinical history?" Unlike many existing methods that focus on static or single-point interventions (Alaa & van der Schaar, 2017; Johansson et al., 2016; Zhang et al., 2020), our method is specifically designed to handle the sequential nature of treatments in medical practice – a feature that is both realistic and necessary, as treatments are rarely applied all at once but rather sequentially over time (Apperloo et al., 2024). With the growing availability of large-scale observational data from EHRs (Allam et al., 2021; Feuerriegel et al., 2024; Bica et al., 2021) and wearable devices (Battalio et al., 2021), there is an increasing need for robust methods that estimate the effect of multiple treatments, given the *individual* patient history.

Our framework builds on three key assumptions: (i) consistency, (ii) positivity, and (iii) sequential ignorability (see Section 3). These assumptions are the *standard* assumptions for estimating CAPOs over time (Bica et al., 2020; Li et al., 2021; Melnychuk et al., 2022; Seedat et al., 2022). Notably, compared to other methods that rely on even *stricter* assumptions, such as additional Markov or independence assumptions (Özyurt et al., 2021), our assumptions are *less* restrictive. Furthermore, these assumptions are the *dynamic* analogues of the standard causal inference assumptions in *static* settings (Alaa & van der Schaar, 2017; Muandet et al., 2021; Johansson et al., 2016). Importantly, methods for the static setting implicitly impose *unrealistic assumption* that treatments occur only once and that covariates and outcomes remain static over time. Such limitations can introduce significant bias in sequential decision-making contexts. In contrast, our approach models the time-varying nature of clinical interventions and patient evolution, making it less restrictive and far more aligned with real-world medical scenarios.

Further, we argue that these assumptions are both plausible and practical in medical applications. First, consistency is generally satisfied as long as EHR data is accurately and systematically recorded. Second, positivity can be ensured through thoughtful data pre-processing, such as filtering observations or applying propensity clipping. Additionally, as the scale of observational datasets grows, this assumption becomes less restrictive. Third, the sequential ignorability assumption is a standard assumption in epidemiology (Little & Rubin, 2000), and studies in digital health interventions may satisfy this assumption by design. Furthermore, advances in sensitivity analysis (Frauen et al., 2023b; Oprescu et al., 2023) and partial identification frameworks (Duarte et al., 2023) offer complementary pathways to relax this assumption. That is, these literature streams are *orthogonal* to our work. In practice, our GT thus integrates into established workflows that include point estimation, uncertainty quantification, and sensitivity analysis.

From a practical perspective, our GT addresses key challenges in estimating CAPOs for sequences of treatments. Specifically, our GT provides a neural end-to-end solution that adjusts for time-varying confounding. On top, it neither relies on large-variance pseudo-outcomes (Prop. 3) nor on estimating high-dimensional probability distributions. Therefore, we are convinced that our GT is an important step towards reliable personalized medicine.

## C DERIVATION OF G-COMPUTATION FOR CAPOS

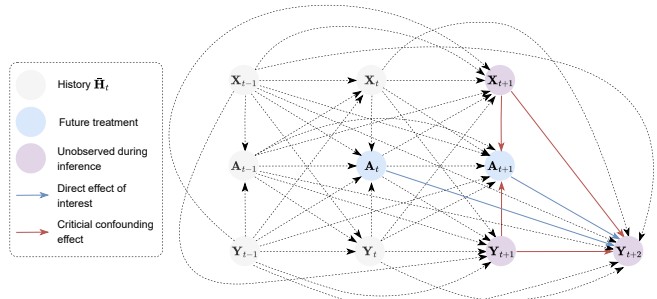

Figure 3: During inference, future time-varying confounders are *unobserved* (here: $(X_{t+1}, Y_{t+1})$). In order to estimate CAPOs for an interventional treatment sequence without **time-varying confounding bias**, proper causal adjustments such as G-computation are required.

In the following, we provide a derivation of the G-computation formula (Bang & Robins, 2005; Robins, 1999; Robins & Hernán, 2009) for CAPOs over time. Recall that G-computation for CAPOs is given by

$$
\mathbb{E}[Y_{t+\tau}[a_{t:t+\tau-1}] \mid \bar{H}_t = \bar{h}_t]
$$

$$
= \mathbb{E}\Big\{ \mathbb{E}\Big[ \ldots \mathbb{E}\big\{ \mathbb{E}[Y_{t+\tau} \mid \bar{H}^t_{t+\tau-1}, A_{t:t+\tau-1} = a_{t:t+\tau-1}] \mid \bar{H}^t_{t+\tau-2}, A_{t:t+\tau-2} = a_{t:t+\tau-2} \big\} \quad (27)
$$

$$
\ldots \Big| \bar{H}^t_{t+1}, A_{t:t+1} = a_{t:t+1} \Big] \Big| \bar{H}_t = \bar{h}_t, A_t = a_t \Big\}.
$$

The following derivation follows the steps in (Frauen et al., 2023a) and extends them to CAPOs:

$$
\mathbb{E}[Y_{t+\tau}[a_{t:t+\tau-1}] \mid \bar{H}_t = \bar{h}_t]
$$

$$
= \mathbb{E}[Y_{t+\tau}[a_{t:t+\tau-1}] \mid \bar{H}_t = \bar{h}_t, A_t = a_t] \quad (28)
$$

$$
= \mathbb{E}[\mathbb{E}\{Y_{t+\tau}[a_{t:t+\tau-1}] \mid \bar{H}^t_{t+1}, A_t = a_t\} \quad (29)
$$

$$
\mid \bar{H}_t = \bar{h}_t, A_t = a_t]
$$

$$
= \mathbb{E}[\mathbb{E}\{Y_{t+\tau}[a_{t:t+\tau-1}] \mid \bar{H}^t_{t+1}, A_{t:t+1} = a_{t:t+1}\} \quad (30)
$$

$$
\mid \bar{H}_t = \bar{h}_t, A_t = a_t]
$$

$$
= \mathbb{E}[\mathbb{E}\{\mathbb{E}[Y_{t+\tau}[a_{t:t+\tau-1}] \mid \bar{H}^t_{t+2}, A_{t:t+1} = a_{t:t+1}] \quad (31)
$$

$$
\mid \bar{H}^t_{t+1}, A_{t:t+1} = a_{t:t+1}\}
$$

$$
\mid \bar{H}_t = \bar{h}_t, A_t = a_t]
$$

$$
= \mathbb{E}[\mathbb{E}\{\mathbb{E}[Y_{t+\tau}[a_{t:t+\tau-1}] \mid \bar{H}^t_{t+2}, A_{t:t+2} = a_{t:t+2}] \quad (32)
$$

$$
\mid \bar{H}^t_{t+1}, A_{t:t+1} = a_{t:t+1}\}
$$

$$
\mid \bar{H}_t = \bar{h}_t, A_t = a_t]
$$

$$
= \ldots
$$

$$
= \mathbb{E}[\ldots \mathbb{E}\{\mathbb{E}[Y_{t+\tau}[a_{t:t+\tau-1}] \mid \bar{H}^t_{t+\tau-1}, A_{t:t+\tau-1} = a_{t:t+\tau-1}] \quad (33)
$$

$$
\mid \bar{H}^t_{t+\tau-2}, A_{t:t+\tau-2} = a_{t:t+\tau-2}\}
$$

$$
\mid \ldots
$$

$$
\mid \bar{H}_t = \bar{h}_t, A_t = a_t]
$$

$$
= \mathbb{E}[\ldots \mathbb{E}\{\mathbb{E}[Y_{t+\tau} \mid \bar{H}^t_{t+\tau-1}, A_{t:t+\tau-1} = a_{t:t+\tau-1}] \quad (34)
$$

$$
\mid \bar{H}^t_{t+\tau-2}, A_{t:t+\tau-2} = a_{t:t+\tau-2}\}
$$

$$
\mid \ldots
$$

$$
\mid \bar{H}_t = \bar{h}_t, A_t = a_t],
$$

where Eq. (28) follows from the positivity and sequential ignorability assumptions, Eq. (29) holds due to the law of total probability, Eq. (30) again follows from the positivity and sequential ignorability assumptions, Eq. (31) is the tower rule, Eq. (32) is again due to the positivity and sequential ignorability assumptions, Eq. (33) follows by iteratively repeating the previous steps, and Eq. (34) follows from the consistency assumption.

# D  REGRESSION-BASED ITERATIVE G-COMPUTATION

## D.1  UNBIASED ESTIMAND

**Proposition 1.** *Our regression-based iterative G-computation yields the CAPO in Eq. (1).*

*Proof.* For the proof, we only need to apply the definition of the pseudo-outcomes $G_{t+\delta}^a$:

$$\mathbb{E}[Y_{t+\tau}[a_{t:t+\tau-1}] \mid \bar{H}_t = \bar{h}_t] \tag{35}$$

$$=\mathbb{E}\Bigg\{\mathbb{E}\Bigg[\dots \mathbb{E}\big\{\mathbb{E}[Y_{t+\tau} \mid \bar{H}_{t+\tau-1}^t, A_{t:t+\tau-1} = a_{t:t+\tau-1}] \mid \bar{H}_{t+\tau-2}^t, A_{t:t+\tau-2} = a_{t:t+\tau-2}\big\}$$

$$\dots \bigg| \bar{H}_{t+1}^t, A_{t:t+1} = a_{t:t+1}\Bigg] \bigg| \bar{H}_t = \bar{h}_t, A_t = a_t\Bigg\} \tag{36}$$

$$=\mathbb{E}\Bigg\{\mathbb{E}\Bigg[\dots \mathbb{E}\big\{\mathbb{E}[G_{t+\tau}^a \mid \bar{H}_{t+\tau-1}^t, A_{t:t+\tau-1} = a_{t:t+\tau-1}] \mid \bar{H}_{t+\tau-2}^t, A_{t:t+\tau-2} = a_{t:t+\tau-2}\big\}$$

$$\dots \bigg| \bar{H}_{t+1}^t, A_{t:t+1} = a_{t:t+1}\Bigg] \bigg| \bar{H}_t = \bar{h}_t, A_t = a_t\Bigg\} \tag{37}$$

$$=\mathbb{E}\Bigg\{\mathbb{E}\Bigg[\dots \mathbb{E}\big\{g_{t+\tau-1}^a(\bar{H}_{t+\tau-1}^t) \mid \bar{H}_{t+\tau-2}^t, A_{t:t+\tau-2} = a_{t:t+\tau-2}\big\}$$

$$\dots \bigg| \bar{H}_{t+1}^t, A_{t:t+1} = a_{t:t+1}\Bigg] \bigg| \bar{H}_t = \bar{h}_t, A_t = a_t\Bigg\} \tag{38}$$

$$=\mathbb{E}\Bigg\{\mathbb{E}\Bigg[\dots \mathbb{E}\big\{G_{t+\tau-1}^a \mid \bar{H}_{t+\tau-2}^t, A_{t:t+\tau-2} = a_{t:t+\tau-2}\big\}$$

$$\dots \bigg| \bar{H}_{t+1}^t, A_{t:t+1} = a_{t:t+1}\Bigg] \bigg| \bar{H}_t = \bar{h}_t, A_t = a_t\Bigg\} \tag{39}$$

$$=\mathbb{E}\Bigg\{\mathbb{E}\Bigg[\dots g_{t+\tau-2}^a(\bar{H}_{t+\tau-2}^t)\dots \bigg| \bar{H}_{t+1}^t, A_{t:t+1} = a_{t:t+1}\Bigg] \bigg| \bar{H}_t = \bar{h}_t, A_t = a_t\Bigg\} \tag{40}$$

$$=\dots \tag{41}$$

$$=\mathbb{E}\Bigg\{G_{t+1}^a \bigg| \bar{H}_t = \bar{h}_t, A_t = a_t\Bigg\} \tag{42}$$

$$=g_t^a(\bar{h}_t), \tag{43}$$

where Eq. (36) holds due the G-computation formula (see Supplement C). $\qquad\square$

## D.2  TARGET OF OUR GT

**Proposition 2.** *Our GT estimates G-computation formula as in and, therefore, performs proper adjustments for time-varying confounders.*

*Proof.* For the proof, we perform the steps as in Supplement D.1:

$$\hat{\mathbb{E}}[Y_{t+\tau}[a_{t:t+\tau-1}] \mid \bar{H}_t = \bar{h}_t] \tag{44}$$

$$=\hat{\mathbb{E}}\Big\{\hat{\mathbb{E}}\Big[\ldots\hat{\mathbb{E}}\big\{\hat{\mathbb{E}}[Y_{t+\tau} \mid \bar{H}^t_{t+\tau-1}, A_{t:t+\tau-1} = a_{t:t+\tau-1}] \mid \bar{H}^t_{t+\tau-2}, A_{t:t+\tau-2} = a_{t:t+\tau-2}\big\}$$

$$\ldots \Big| \bar{H}^t_{t+1}, A_{t:t+1} = a_{t:t+1}\Big] \Big| \bar{H}_t = \bar{h}_t, A_t = a_t\Big\} \tag{45}$$

$$=\hat{\mathbb{E}}\Big\{\hat{\mathbb{E}}\Big[\ldots\hat{\mathbb{E}}\big\{\hat{\mathbb{E}}[\tilde{G}^a_{t+\tau} \mid \bar{H}^t_{t+\tau-1}, A_{t:t+\tau-1} = a_{t:t+\tau-1}] \mid \bar{H}^t_{t+\tau-2}, A_{t:t+\tau-2} = a_{t:t+\tau-2}\big\}$$

$$\ldots \Big| \bar{H}^t_{t+1}, A_{t:t+1} = a_{t:t+1}\Big] \Big| \bar{H}_t = \bar{h}_t, A_t = a_t\Big\} \tag{46}$$

$$=\hat{\mathbb{E}}\Big\{\hat{\mathbb{E}}\Big[\ldots\hat{\mathbb{E}}\big\{g^{\tau-1}_\phi(a_{t+\tau-1}, z_\theta(\bar{H}_{t+\tau-1}, a_{t:t+\tau-2})) \mid \bar{H}^t_{t+\tau-2}, A_{t:t+\tau-2} = a_{t:t+\tau-2}\big\}$$

$$\ldots \Big| \bar{H}^t_{t+1}, A_{t:t+1} = a_{t:t+1}\Big] \Big| \bar{H}_t = \bar{h}_t, A_t = a_t\Big\} \tag{47}$$

$$=\hat{\mathbb{E}}\Big\{\hat{\mathbb{E}}\Big[\ldots\hat{\mathbb{E}}\big\{\tilde{G}^a_{t+\tau-1} \mid \bar{H}^t_{t+\tau-2}, A_{t:t+\tau-2} = a_{t:t+\tau-2}\big\}$$

$$\ldots \Big| \bar{H}^t_{t+1}, A_{t:t+1} = a_{t:t+1}\Big] \Big| \bar{H}_t = \bar{h}_t, A_t = a_t\Big\} \tag{48}$$

$$=\hat{\mathbb{E}}\Big\{\hat{\mathbb{E}}\Big[\ldots g^{\tau-2}_\phi(a_{t+\tau-2}, z_\theta(\bar{H}_{t+\tau-2}, a_{t:t+\tau-3}))\ldots \Big| \bar{H}^t_{t+1}, A_{t:t+1} = a_{t:t+1}\Big] \Big| \bar{H}_t = \bar{h}_t, A_t = a_t\Big\} \tag{49}$$

$$=\ldots \tag{50}$$

$$=\hat{\mathbb{E}}\Big\{\tilde{G}^a_{t+1} \Big| \bar{H}_t = \bar{h}_t, A_t = a_t\Big\} \tag{51}$$

$$=g^0_\phi(a_t, z_\theta(\bar{h}_t)). \tag{52}$$

$\square$

### D.3 EXAMPLES

To illustrate how regression-based iterative G-computation works, we apply the procedure to two examples. First, we show the trivial case for $(\tau = 1)$-step-ahead predictions and, then, for $(\tau = 2)$-step-ahead predictions. Recall that the following only holds under our standard assumptions (i) *consistency*, (ii) *positivity*, and (iii) *sequential ignorability*.

$(\tau = 1)$-step-ahead prediction:

This is the trivial case, as there is *no time-varying confounding*. Instead, all confounders are observed in the history. Therefore, we can simply condition on the observed history and resemble the *backdoor-adjustment* from the static setting. Importantly, this is **not** the focus of our work, but we show it for illustrative purposes:

$$\mathbb{E}\big[Y_{t+1}[a_t] \mid \bar{H}_t = \bar{h}_t\big] \tag{53}$$

$$\underbrace{=}_{\text{Ass. (ii)+(iii)}} \mathbb{E}\big[Y_{t+1}[a_t] \mid \bar{H}_t = \bar{h}_t, A_t = a_t\big] \tag{54}$$

$$\underbrace{=}_{\text{Ass. (i)}} \mathbb{E}\big[Y_{t+1} \mid \bar{H}_t = \bar{h}_t, A_t = a_t\big] \tag{55}$$

$$\underbrace{=}_{\text{Def. } G_{t+1}^a} \mathbb{E}\big[G_{t+1}^a \mid \bar{H}_t = \bar{h}_t, A_t = a_t\big] \tag{56}$$

$$\underbrace{=}_{\text{Def. } g_t^a} g_t^a(\bar{h}_t). \tag{57}$$

$(\tau = 2)$-step-ahead prediction:

$(\tau = 2)$-step-ahead predictions already incorporate all the difficulties that are present for multi-step ahead predictions. Here, we need to account for future time-varying confounders such as $(X_{t+1}, Y_{t+1})$ as in Figure 3:

$$\mathbb{E}\big[Y_{t+2}[a_{t:t+1}] \mid \bar{H}_t = \bar{h}_t\big] \tag{58}$$

$$\underbrace{=}_{\text{Ass. (ii)+(iii)}} \mathbb{E}\big[Y_{t+2}[a_{t:t+1}] \mid \bar{H}_t = \bar{h}_t, A_t = a_t\big] \tag{59}$$

$$\underbrace{=}_{\text{Law of total prob.}} \mathbb{E}\Big[\mathbb{E}\big[Y_{t+2}[a_{t:t+1}] \mid \bar{H}_{t+1}^t, A_t = a_t\big] \mid \bar{H}_t = \bar{h}_t, A_t = a_t\Big] \tag{60}$$

$$\underbrace{=}_{\text{Ass. (ii)+(iii)}} \mathbb{E}\Big[\mathbb{E}\big[Y_{t+2}[a_{t:t+1}] \mid \bar{H}_{t+1}^t, A_{t:t+1} = a_{t:t+1}\big] \mid \bar{H}_t = \bar{h}_t, A_t = a_t\Big] \tag{61}$$

$$\underbrace{=}_{\text{Ass. (i)}} \mathbb{E}\Big[\mathbb{E}\big[Y_{t+2} \mid \bar{H}_{t+1}^t, A_{t:t+1} = a_{t:t+1}\big] \mid \bar{H}_t = \bar{h}_t, A_t = a_t\Big] \tag{62}$$

$$\underbrace{=}_{\text{Def. } G_{t+2}^a} \mathbb{E}\Big[\mathbb{E}\big[G_{t+2}^a \mid \bar{H}_{t+1}^t, A_{t:t+1} = a_{t:t+1}\big] \mid \bar{H}_t = \bar{h}_t, A_t = a_t\Big] \tag{63}$$

$$\underbrace{=}_{\text{Def. } g_{t+1}^a} \mathbb{E}\big[g_{t+1}^a(\bar{H}_{t+1}^t) \mid \bar{H}_t = \bar{h}_t, A_t = a_t\big] \tag{64}$$

$$\underbrace{=}_{\text{Def. } G_{t+1}^a} = \mathbb{E}\big[G_{t+1}^a \mid \bar{H}_t = \bar{h}_t, A_t = a_t\big] \tag{65}$$

$$\underbrace{=}_{\text{Def. } g_t^a} g_t^a(\bar{h}_t). \tag{66}$$

# E VARIANCE OF INVERSE PROPENSITY WEIGHTING

In this section, we compare two possible approaches to adjust for time-varying confounders: G-computation and inverse propensity weighting (IPW) (Robins & Hernán, 2009; Robins et al., 2000), which is leveraged by RMSNs (Lim et al., 2018).

For a fair comparison of G-computation and IPW, we compare the *variance of the ground-truth pseudo-outcomes* that each method relies on – that is, the $G_{t+\delta}^a$ of our GT and the inverse propensity weighted outcomes of RMSNs. Importantly, a larger variance of the pseudo-outcomes will directly translate into a larger variance of the respective estimator. We find that IPW leads to a larger variance, which is why we prefer G-computation in our GT.

**Proposition 3.** *Pseudo-outcomes constructed via inverse propensity weighting have larger variance than pseudo-outcomes in our G-transformer.*

*Proof.* To simplify notation, we consider the variance of the pseudo-outcomes in the *static setting*. The analog directly translates into the time-varying setting.

Let $Y$ be the outcome, $X$ the covariates, and $A$ the treatment. Without loss of generality, we consider the potential outcome for $A = 1$.

For G-computation, the variance of the pseudo-outcome $g^1(X)$ is given by

$$\text{Var}[g^1(X)] = \text{Var}[\mathbb{E}[Y \mid X, A = 1]] \tag{67}$$

$$= \mathbb{E}\Big[\mathbb{E}[Y \mid X, A = 1]^2\Big] - \mathbb{E}\Big[\mathbb{E}[Y \mid X, A = 1]\Big]^2 \tag{68}$$

$$= \mathbb{E}\Big[\mathbb{E}[Y \mid X, A = 1]^2\Big] - \mathbb{E}\Big[Y[1]\Big]^2. \tag{69}$$

For IPW, the variance of the pseudo-outcome is

$$\text{Var}\Big[\frac{YA}{\pi(X)}\Big] = \mathbb{E}\Big[\Big(\frac{YA}{\pi(X)}\Big)^2\Big] - \mathbb{E}\Big[\frac{YA}{\pi(X)}\Big]^2 \tag{70}$$

$$= \mathbb{E}\Big[\mathbb{E}\Big[\frac{Y^2A}{\pi^2(X)} \mid X\Big]\Big] - \mathbb{E}\Big[Y[1]\Big]^2 \tag{71}$$

$$= \mathbb{E}\Big[\mathbb{E}\Big[\frac{Y^2\pi(X)}{\pi^2(X)} \mid X, A = 1\Big]\Big] - \mathbb{E}\Big[Y[1]\Big]^2 \tag{72}$$

$$= \mathbb{E}\Big[\underbrace{\frac{1}{\pi(X)}}_{\geq 1} \mathbb{E}[Y^2 \mid X, A = 1]\Big] - \mathbb{E}\Big[Y[1]\Big]^2, \tag{73}$$

and, with

$$\mathbb{E}[Y \mid X, A = 1]^2 + \underbrace{\text{Var}[Y \mid X, A = 1]}_{\geq 0} = \mathbb{E}[Y^2 \mid X, A = 1], \tag{74}$$

we have that

$$\text{Var}\Big[\frac{YA}{\pi(X)}\Big] \geq \text{Var}[g^1(X)]. \tag{75}$$

Therefore, we conclude that G-computation leads to a lower variance than IPW and, hence, our GT has a lower variance than RMSNs. □

**Remarks:**

- The inverse propensity weight is what really drives the difference in variance between the approaches. Note that, in the time-varying setting, IPW relies on *products of inverse propensities*, which can lead to even more extreme weights for multi-step ahead predictions.
- IPW is particularly problematic when there are overlap violations in the data. However, as the input history $\bar{H}_t$ in the time-varying setting is very high-dimensional (i.e., $t \times (d_x + d_y)$-dimensional), overlap violations are even more problematic. This is another advantage for our method.

## F    COMPARISON TO G-NET

In this section, we compare our iterative regression-based approach to G-computation to the version that is employed by G-Net (Li et al., 2021).

G-Net makes a Monte Carlo approximation of Eq. (3) through

$$\int_{\mathbb{R}^{d_x \times \tau - 1} \times \mathbb{R}^{d_y \times \tau - 1}} \mathbb{E}[Y_{t+\tau} \mid \bar{H}_{t+\tau-1}^t = \bar{h}_{t+\tau-1}^t, A_{t:t+\tau-1} = a_{t:t+\tau-1}]$$
$$\times \prod_{\delta=1}^{\tau-1} p(x_{t+\delta}, y_{t+\delta} \mid \bar{h}_t, x_{t+1:t+\delta-1}, y_{t+1:t+\delta-1}, a_{t:t+\delta-1}) \, d(x_{t+1:t+\tau-1}, y_{t+1:t+\tau-1}). \quad (76)$$

For this, G-Net requires estimating the full distribution

$$\prod_{\delta=1}^{\tau-1} dp(x_{t+\delta}, y_{t+\delta} \mid \bar{h}_t, x_{t+1:t+\delta-1}, y_{t+1:t+\delta-1}, a_{t:t+\delta-1}). \quad (77)$$

That is, for $\tau$-step ahead predictions, G-Net estimates a $(\tau - 1) \times (d_x + d_y)$-dimensional probability distribution.

We compare the approach of G-Net to to our regression-based G-computation in Table 4.

| Estimated moment | | 1st | 2nd | 3rd | 4th | ... | $\infty$ |
|---|---|---|---|---|---|---|---|
| **Dimension** | G-Net (Li et al., 2021) | $(\tau-1) \times (d_x+d_y) + d_y$ | $(\tau-1) \times (d_x+d_y)$ | $(\tau-1) \times (d_x+d_y)$ | $(\tau-1) \times (d_x+d_y)$ | ... | $(\tau-1) \times (d_x+d_y)$ |
| | GT (ours) | $\tau \times d_y$ | – | – | – | ... | – |

Table 4: We compare the approach to G-computation of G-Net (Li et al., 2021) to our regression-based version. For this, we compare the *dimensions of the estimated moments* for each method, respectively. G-Net requires estimating the full distribution of all time-varying confounders in the future. This means that **all moments** of **all time-varying confounders** at **all time steps in the future** need to be estimated. In contrast, our GT *only requires estimation of the first moment of the lower-dimensional target variable*, which is a clear advantage.

# G    ADDITIONAL RESULTS

## G.1    ADDITIONAL RESULTS AND ABLATIONS

In the following, we report the performance of two ablations: the **(A) G-LSTM** and the **(B) biased transformer (BT)**. For this, we show **(C) additional results** of our GT, the baselines, and the two ablations.

**(A) G-LSTM:** Our first ablation is the G-LSTM. For this, we replaced the transformer backbone $z_\theta(\cdot)$ of our GT by an LSTM network. We find that our **G-LSTM is highly effective**: it outperforms all baselines from the literature while our proposed G-transformer is still superior. This demonstrates that our novel method for iterative regression-based G-computation is both effective and general.

**(B) BT:** Additionally, we implement a biased transformer (BT). Here, we leverage the same transformer backbone $z_\theta(\cdot)$ as in our GT, but we directly train the output heads on the factual data. Thereby, the BT refrains from performing G-computation. We can thus isolate the contribution of the iterative G-computation to the overall performance. Our results show that the **BT suffers from significant estimation bias** and, therefore, demonstrates that our proper adjustments for time-varying confounders are required for accurate estimates of CAPOs.

**(C) Additional results:** We report additional results on both (i) fully synthetic data as in Section 5.1 and on (ii) semi-synthetic data as in Section 5.2.

For (i) fully synthetic data, we report the performance of all methods for lower levels of confounding in Figure 4 and additional prediction windows up to $\tau = 6$ for fixed level of confounding $\gamma = 10.0$ in Figure 5.

For (ii) semi-synthetic data, we report additional prediction windows up to $\tau = 12$ for $N = 1000$ in Figure 6.

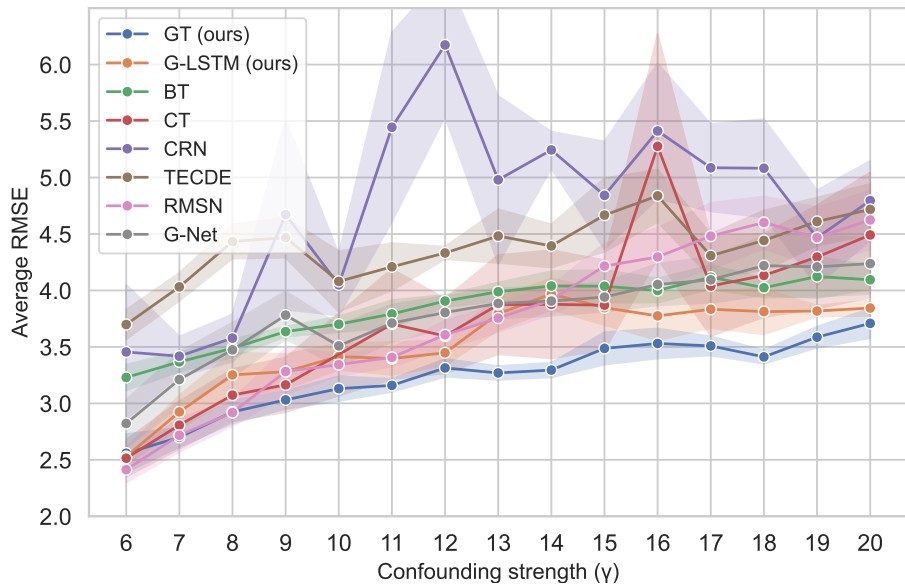

Figure 4: Synthetic data: We **decrease the confounding strength** ($\gamma = 6, 7, 8, 9$) for $\tau = 2$. Additionally, we report previous results of the baselines with the **new ablations: G-LSTM and BT**. Notably, our G-LSTM has competitive performance, while BT suffers from significant bias. Our *GT remains the strongest method*. We see a similar picture as for Figure 5 and Figure 6: our methods perform the best due to our novel, iterative G-computation.

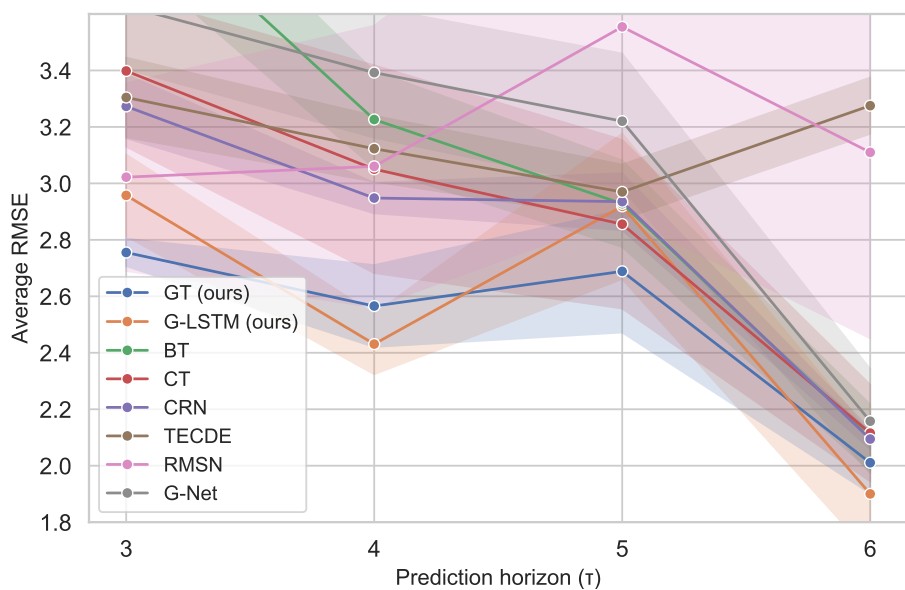

Figure 5: Synthetic data: We **increase the prediction horizon** up to $\tau = 6$ for confounding $\gamma = 10$. Our G-LSTM and our GT have the overall *best performance on all prediction windows*. The results coincide with our results in Figure 4 and Figure 6; our approach to G-computation leads to the lowest prediction errors. (Please note that decreasing prediction errors for increasing $\tau$ is due to the strong heteroscedasticity of the outcome variable; smaller $\tau$ means that we predict more samples in the test data for very small $t$, where variance is the highest.)

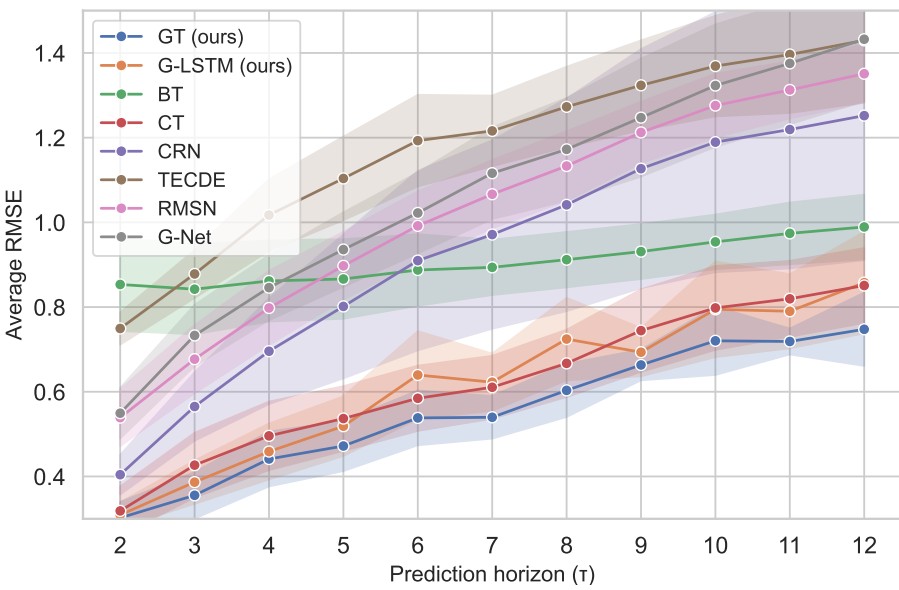

Figure 6: Semi-synthetic data: We **increase the prediction horizon** up to $\tau = 12$ for $N = 1000$ training samples. We further **implement two ablations**: our G-LSTM and the biased transformer (BT). As in Figure 4 and Figure 5, our G-LSTM almost consistently outperforms the baselines, while the BT has large errors. Our *GT remains the best for all prediction windows*. This shows that our novel approach for G-computation leads to accurate predictions, irrespective of the neural backbone. Further, it shows that proper adjustments are important for CAPO estimation.

## G.2 SENSITIVITY TO NOISE IN PSEUDO-OUTCOMES

Finally, we provide more insights into the quality of the generated pseudo-outcomes $\tilde{G}^a_{t+\delta}$ in Figure 7. Here, we added increasing levels of constant bias to the pseudo-outcomes during training. Our results show that these artificial corruptions indeed lead to a significant decrease in the overall performance of our GT. We therefore conclude that, without artificial corruption, our generated pseudo-outcomes are good estimates of the true nested expectations. Further, this shows that correct estimates of the pseudo-outcomes are indeed necessary for high-quality unbiased estimates. Of note, the quality of the predicted pseudo-outcomes is also directly validated by the strong empirical performance in Section 5.

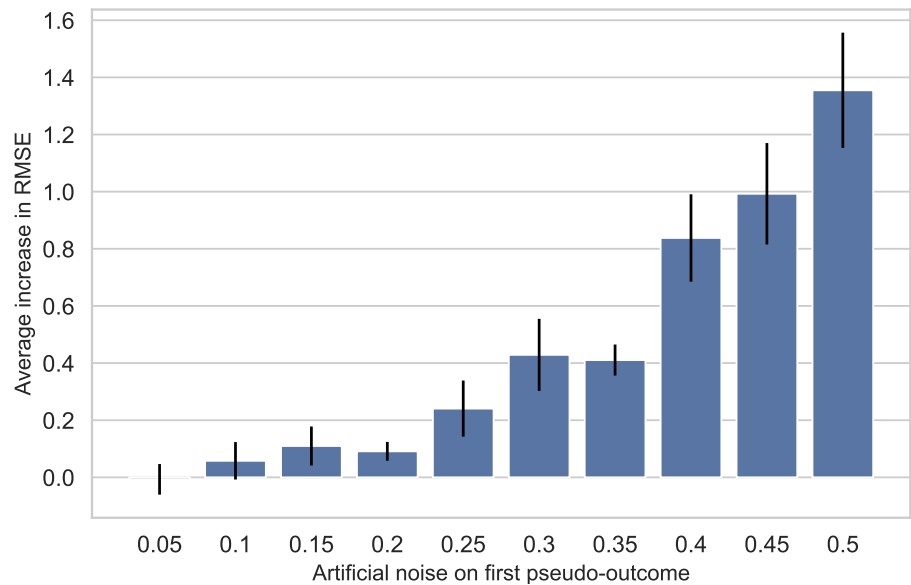

Figure 7: During training, we add **artificial levels of noise to the pseudo-outcomes** of our GT (prediction window $\tau = 2$, confounding strength $\gamma = 10$ on synthetic data). We see that performance quickly deteriorates. This is expected, as it implies that the pseudo-outcomes generated by our GT are meaningful and important for accurate, unbiased predictions.

# H  EXPERIMENTS ON REAL-WORLD DATA

In this section, we empirically demonstrate that our method performs well for predicting patient outcomes on factual data. Importantly, predicting *factual outcomes* is **not** what our GT is primarily designed for. In particular, any standard regression model suffices for this task, and **no** additional adjustments are required to account for time-varying confounding. Instead, our GT is trained to estimate CAPOs, which is a counterfactual quantity in the time-varying setting.

We use the MIMIC-III dataset (Johnson et al., 2016; Wang et al., 2020), which gives measurements from intensive care units aggregated at hourly levels. Here, we predict the effect of vasopressors and mechanical ventilation on diastolic blood pressure. Our setup closely follows (Melnychuk et al., 2022), and we additionally vary our sample size for training. The results are reported in Figure 8. We find that *our GT performs best even for real-world prediction tasks* although this task does **not** require adjustments. This demonstrates that our method is directly applicable to predict real-world patient outcomes. Further, it shows that the way we adjust does **not** deteriorate performance when there is nothing to adjust and, thus, is highly effective.

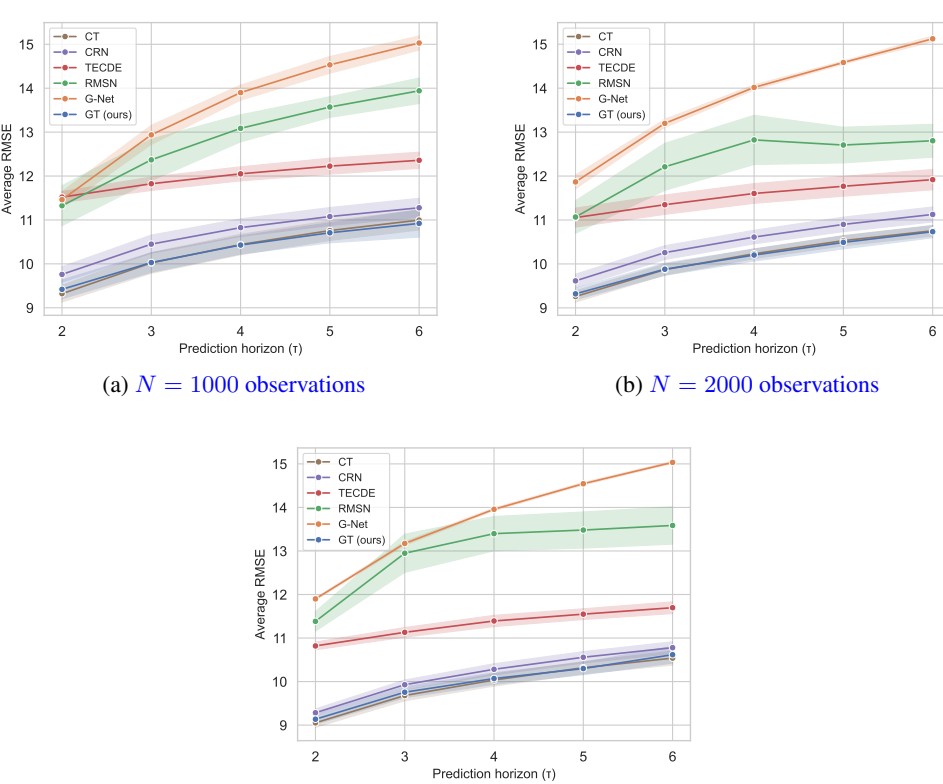

(a) $N = 1000$ observations

(b) $N = 2000$ observations

(c) $N = 3000$ observations

Figure 8: **Performance for real-world data.** We evaluate our GT and the baselines on real-world data. We use the MIMIC-III dataset (Johnson et al., 2016) and report the RMSE for predicting the effect of vasopressors and mechanical ventilation on diastolic blood pressure. Our GT performs best along with CT (Melnychuk et al., 2022), followed by CRN (Bica et al., 2020). This is expected, as evaluation on factual data does **not** require adjustments for time-varying confounding. Importantly, we can see that our iterative regression approach leads to very accurate prediction results even on factual data. This further underlines that our GT is directly applicable to medical datasets.

# I  COEFFICIENT OF VARIATION

In the following, we additionally report the coefficient of variation of our main study in Section 5. Lower values in the coefficient of variation indicate more stable predictions. Table 5 shows the results. Clearly, our GT is superior to the baselines and has significantly more robust estimates of the CAPO.

| | $\gamma = 10$ | $\gamma = 11$ | $\gamma = 12$ | $\gamma = 13$ | $\gamma = 14$ | $\gamma = 15$ | $\gamma = 16$ | $\gamma = 17$ | $\gamma = 18$ | $\gamma = 19$ | $\gamma = 20$ |
|---|---|---|---|---|---|---|---|---|---|---|---|
| CRN (Bica et al., 2020) | 0.14 | 0.31 | 0.21 | 0.30 | 0.06 | 0.20 | 0.22 | 0.15 | 0.17 | 0.19 | 0.15 |
| TE-CDE (Seedat et al., 2022) | 0.13 | 0.10 | 0.03 | 0.10 | 0.09 | 0.14 | 0.10 | 0.09 | 0.12 | 0.09 | 0.10 |
| CT (Melnychuk et al., 2022) | 0.21 | 0.21 | 0.17 | 0.18 | 0.19 | 0.17 | 0.32 | 0.18 | 0.22 | 0.17 | 0.21 |
| RMSNs (Lim et al., 2018) | **0.06** | 0.05 | 0.07 | 0.07 | 0.07 | 0.09 | 0.12 | 0.13 | 0.10 | 0.12 | 0.11 |
| G-Net (Li et al., 2021) | 0.11 | 0.09 | 0.08 | 0.07 | 0.07 | **0.07** | 0.09 | 0.10 | 0.13 | 0.13 | 0.11 |
| **GT** (ours) | 0.07 | **0.04** | **0.06** | **0.04** | **0.03** | 0.09 | **0.07** | **0.07** | **0.09** | **0.06** | **0.07** |

Table 5: Coefficient of variation on synthetic data based on the tumor growth model with $\tau = 2$. Lower values indicate more stable predictions. Our GT clearly outperforms the baselines.

## J  ARCHITECTURE OF G-TRANSFORMER

In the following, we provide details on the architecture of our GT.

**Multi-input transformer:** The multi-input transformer as the backbone of our GT is motivated by (Melnychuk et al., 2022), which develops an architecture that is tailored for the types of data that are typically available in medical scenarios: (i) outcomes $\bar{Y}_t \in \mathbb{R}^{d_y \times t}$, covariates $\bar{X}_t \in \mathbb{R}^{d_x \times t}$, and treatments $\bar{A}_t \in \{0,1\}^{d_a \times t}$. In particular, their proposed transformer model consists of three separate sub-transformers, where each sub-transformer performs *multi-headed self-attention mechanisms* on one particular data input. Further, these sub-transformers are connected with each other through *in-between cross-attention mechanisms*, ensuring that information is exchanged. Therefore, we build on this idea as the backbone of our GT, as we detail below.

Our multi-input transformer $z_\theta(\cdot)$ consists of three sub-transformer models $z_\theta^k(\cdot)$, $k = 1, 2, 3$, where $z_\theta^k(\cdot)$ focuses on one data input $\bar{U}_t^k \in \{\bar{Y}_t, \bar{X}_t, \bar{A}_{t-1}\}$, $k \in \{1, 2, 3\}$, respectively.

(1) Input transformations: First, the data $\bar{U}_t^k \in \mathbb{R}^{d_k \times t}$ is linearly transformed through

$$Z_t^{k,0} = (\bar{U}_t^k)^\top W^{k,0} + b^{k,0} \in \mathbb{R}^{t \times d_h} \tag{78}$$

where $W^{k,0} \in \mathbb{R}^{d_k \times d_h}$ and $b^{k,0} \in \mathbb{R}^{d_h}$ are the weight matrix and the bias, respectively, and $d_h$ is the number of transformer units.

(2) Transformer blocks: Next, we stack $j = 1, \ldots, J$ transformer blocks, where each transformer block $j$ receives the outputs $Z_t^{k,j-1}$ of the previous transformer block $j-1$. For this, we combine (i) *multi-headed self- and cross-attentions*, and (ii) *feed-forward networks*.

(i) *Multi-headed self- and cross-attentions:* The output of block $j$ for sub-transformer $k$ is given by the *multi-headed cross-attention*

$$Z_t^{k,j} = \tilde{Q}_t^{k,j} + \sum_{l \neq k} \text{MHA}(\tilde{Q}_t^{k,j}, \tilde{K}_t^{l,j}, \tilde{V}_t^{l,j}), \tag{79}$$

where $\tilde{Q}_t^{k,j} = \tilde{K}_t^{k,j} = \tilde{V}_t^{k,j}$ are the outputs of the *multi-headed self-attentions*

$$\tilde{Q}_t^{k,j} = Z_t^{k,j-1} + \text{MHA}(Q_t^{k,j}, K_t^{k,j}, V_t^{k,j}). \tag{80}$$

Here, MHA$(\cdot)$ denotes the multi-headed attention mechanism as in (Vaswani et al., 2017) given by

$$\text{MHA}(q, k, v) = (\text{Attention}(q^1, k^1, v^1), \ldots, \text{Attention}(q^M, k^M, v^M)), \tag{81}$$

where

$$\text{Attention}(q^m, k^m, v^m) = \text{softmax}\left(\frac{q^m (k^m)^\top}{\sqrt{d_{qkv}}}\right) v^m \tag{82}$$

is the attention mechanism for $m = 1, \ldots, M$ attention heads. The queries, keys, and values $q^m, k^m, v^m \in \mathbb{R}^{t \times d_{qkv}}$ have dimension $d_{qkv}$, which is equal to the hidden size $d_h$ divided by the number of attention heads $M$, that is, $d_{qkv} = d_h / M$. For this, we compute the queries, keys, and values for the *cross-attentions* as

$$\tilde{Q}_t^{k,m,j} = \tilde{Q}_t^{k,j} \tilde{W}^{k,m,j} + \tilde{b}^{k,m,j} \in \mathbb{R}^{t \times d_{qkv}}, \tag{83}$$

$$\tilde{K}_t^{k,m,j} = \tilde{K}_t^{k,j} \tilde{W}^{k,m,j} + \tilde{b}^{k,m,j} \in \mathbb{R}^{t \times d_{qkv}}, \tag{84}$$

$$\tilde{V}_t^{k,m,j} = \tilde{V}_t^{k,j} \tilde{W}^{k,m,j} + \tilde{b}^{k,m,j} \in \mathbb{R}^{t \times d_{qkv}}, \tag{85}$$

and for the *self-attentions* as

$$Q_t^{k,m,j} = Z_t^{k,j-1} W^{k,m,j} + b^{k,m,j} \in \mathbb{R}^{t \times d_{qkv}}, \tag{86}$$

$$K_t^{k,m,j} = Z_t^{k,j-1} W^{k,m,j} + b^{k,m,j} \in \mathbb{R}^{t \times d_{qkv}}, \tag{87}$$

$$V_t^{k,m,j} = Z_t^{k,j-1} W^{k,m,j} + b^{k,m,j} \in \mathbb{R}^{t \times d_{qkv}}. \tag{88}$$

where $\tilde{W}^{k,m,j}, W^{k,m,j} \in \mathbb{R}^{d_h \times d_{qkv}}$ and $b^{k,m,j}, \tilde{b}^{k,m,j} \in \mathbb{R}^{d_q kv}$ are the trainable weights and biases for sub-transformers $k = 1, 2, 3$, transformer blocks $j = 1, \ldots, J$, and attention heads

$m = 1, \ldots, M$. Of note, each *self- and cross attention* uses relative positional encodings (Shaw et al., 2018) to preserve the order of the input sequence as in (Melnychuk et al., 2022).

(ii) *Feed-forward networks:* After the *multi-headed cross-attention* mechanism, our GT applies a feed-forward neural network on each $Z_t^{k,j}$, respectively. Further, we apply dropout and layer normalizations (Ba et al., 2016) as in (Melnychuk et al., 2022; Vaswani et al., 2017). That is, our GT transforms the output $Z_t^{k,j}$ for transformer block $j$ of sub-transformer $k$ through a sequence of transformations

$$\text{FF}^{k,j}(Z_t^{k,j}) = \text{LayerNorm} \circ \text{Dropout} \circ \text{Linear} \circ \text{Dropout} \circ \text{ReLU} \circ \text{Linear}(Z_t^{k,j}). \quad (89)$$

(3) Output transformation: Finally, after transformer block $J$, we apply a final transformation with dropout and average the outputs as

$$Z_t^A = \text{ELU} \circ \text{Linear} \circ \text{Dropout}\left(\frac{1}{3} \sum_{k=1}^{3} Z_t^{k,J}\right), \quad (90)$$

such that $Z_t^A \in \mathbb{R}^{d_z}$

**G-computation heads:** The *G-computation heads* $\{g_\phi^\delta(\cdot)\}_{\delta=0}^{\tau-1}$ receive the corresponding hidden state $Z_{t+\delta}^A$ and the current treatment $A_{t+\delta}$ and transform it with another feed-forward network through

$$g_\phi^\delta(Z_{t+\delta}^A, A_{t+\delta}) = \text{Linear} \circ \text{ELU} \circ \text{Linear}(Z_{t+\delta}^A, A_{t+\delta}). \quad (91)$$

## K ALGORITHMS FOR ITERATIVE TRAINING AND INFERENCE TIME

In Algorithm 1, we summarize the iterative training procedure of our GT and how inference is achieved.

---

**Algorithm 1:** Training and inference with GT.

---

**Training:**

**Input** : Data $\bar{H}_{T-1}, A_{T-1}, Y_T$, treatment sequence $a \in \{0,1\}^{d_a \times \tau}$, learning rate $\eta$
**Output :** Trained GT networks $z_\theta, \{g_\phi^\delta\}_{\delta=0}^{\tau-1}$

**for** $t = 1, \ldots, T - \tau$ **do**
  // Initialize
  $a_{t:t+\tau-1} \hookleftarrow a$
  $\tilde{G}_{t+\tau}^a \hookleftarrow Y_{t+\tau}$
  // Ⓐ Generation step
  **for** $\delta = 1, \ldots, \tau - 1$ **do**
    $Z_{t+\delta}^a \hookleftarrow z_\theta(\bar{H}_{t+\delta}^t, a_{t:t+\delta-1})$
    $\tilde{G}_{t+\delta}^a \hookleftarrow g_\phi^\delta(Z_{t+\delta}^a, a_{t+\delta})$
  **end**

  // Ⓑ Learning step
  **for** $\delta = 0, \ldots, \tau - 1$ **do**
    $Z_{t+\delta}^A \leftarrow z_\theta(\bar{H}_{t+\delta})$
    $\mathcal{L}_t^\delta \leftarrow \left( g_\phi^\delta(Z_{t+\delta}^A, A_{t+\delta}) - \tilde{G}_{t+\delta+1}^a \right)^2$
  **end**
**end**
// Compute gradient and update GT parameters $\phi$
$\phi \leftarrow \phi - \eta \nabla_\phi \left( \frac{1}{T-\tau} \sum_{t=1}^{T-\tau} \left( \frac{1}{\tau} \sum_{\delta=0}^{\tau-1} \mathcal{L}_t^\delta \right) \right)$

**Inference:**

**Input** : Data $\bar{H}_t = \bar{h}_t$, treatment sequence $a \in \{0,1\}^{d_a \times \tau}$
**Output :** $\tilde{g}_t^a = \hat{\mathbb{E}}[G_{t+1}^a \mid \bar{H}_t = \bar{h}_t, a_t]$

// Initialize
$a_{t:t+\tau-1} \hookleftarrow a$
// Ⓐ Generation step
$\hat{g}_t^a \hookleftarrow g_\phi^0(z_\theta(\bar{H}_t), a_t)$

---

Legend: Operations with "←" are attached to the computational graph, while operations with "↩" are detached from the computational graph.

## L    IMPLEMENTATION DETAILS

In Supplements L.1 and L.2, we report details on the hyperparameter tuning. Here, we ensure that the total number of weights is comparable for each method and choose the grids accordingly. All methods are tuned on the validation datasets. As the validation sets only consist of *observational data* instead of interventional data, we tune all methods for $\tau = 1$-step ahead predictions as in (Melnychuk et al., 2022). All methods were optimized with Adam (Kingma & Ba, 2015). Further, we perform a random grid search as in (Melnychuk et al., 2022).

On average, training our GT on fully synthetic data took 13.7 minutes. Further, training on semi-synthetic data with $N = 1000/2000/3000$ samples took $1.1/2.1/3.0$ hours. This is comparable to the baselines. All methods were trained on $1\times$ NVIDIA A100-PCIE-40GB. Overall, running our experiments took approximately 7 days (including hyperparameter tuning).

## L.1 HYPERPARAMETER TUNING: SYNTHETIC DATA

| Method | Component | Hyperparameter | Tuning range |
|---|---|---|---|
| CRN (Bica et al., 2020) | Encoder | LSTM layers ($J$) | 1 |
| | | Learning rate ($\eta$) | 0.01, 0.001, 0.0001 |
| | | Minibatch size | 64, 128, 256 |
| | | LSTM hidden units ($d_h$) | $0.5d_{yxa}, 1d_{yxa}, 2d_{yxa}, 3d_{yxa}, 4d_{yxa}$ |
| | | Balanced representation size ($d_z$) | $0.5d_{yxa}, 1d_{yxa}, 2d_{yxa}, 3d_{yxa}, 4d_{yxa}$ |
| | | FC hidden units ($n_{FC}$) | $0.5d_z, 1d_z, 2d_z, 3d_z, 4d_z$ |
| | | LSTM dropout rate ($p$) | 0.1, 0.2 |
| | | Number of epochs ($n_e$) | 50 |
| | Decoder | LSTM layers ($J$) | 1 |
| | | Learning rate ($\eta$) | 0.01, 0.001, 0.0001 |
| | | Minibatch size | 256, 512, 1024 |
| | | LSTM hidden units ($d_h$) | Balanced representation size of encoder |
| | | Balanced representation size ($d_z$) | $0.5d_{yxa}, 1d_{yxa}, 2d_{yxa}, 3d_{yxa}, 4d_{yxa}$ |
| | | FC hidden units ($n_{FF}$) | $0.5d_z, 1d_z, 2d_z, 3d_z, 4d_z$ |
| | | LSTM dropout rate ($p$) | 0.1, 0.2 |
| | | Number of epochs ($n_e$) | 50 |
| TE-CDE (Seedat et al., 2022) | Encoder | Neural CDE (Kidger et al., 2020) hidden layers ($J$) | 1 |
| | | Learning rate ($\eta$) | 0.01, 0.001, 0.0001 |
| | | Minibatch size | 64, 128, 256 |
| | | Neural CDE hidden units ($d_h$) | $0.5d_{yxa}, 1d_{yxa}, 2d_{yxa}, 3d_{yxa}, 4d_{yxa}$ |
| | | Balanced representation size ($d_z$) | $0.5d_{yxa}, 1d_{yxa}, 2d_{yxa}, 3d_{yxa}, 4d_{yxa}$ |
| | | Feed-forward hidden units ($n_{FF}$) | $0.5d_z, 1d_z, 2d_z, 3d_z, 4d_z$ |
| | | Neural CDE dropout rate ($p$) | 0.1, 0.2 |
| | | Number of epochs ($n_e$) | 50 |
| | Decoder | Neural CDE hidden layers ($J$) | 1 |
| | | Learning rate ($\eta$) | 0.01, 0.001, 0.0001 |
| | | Minibatch size | 256, 512, 1024 |
| | | Neural CDE hidden units ($d_h$) | Balanced representation size of encoder |
| | | Balanced representation size ($d_z$) | $0.5d_{yxa}, 1d_{yxa}, 2d_{yxa}, 3d_{yxa}, 4d_{yxa}$ |
| | | Feed-forward hidden units ($n_{FF}$) | $0.5d_z, 1d_z, 2d_z, 3d_z, 4d_z$ |
| | | Neural CDE dropout rate ($p$) | 0.1, 0.2 |
| | | Number of epochs ($n_e$) | 50 |
| CT (Melnychuk et al., 2022) | (end-to-end) | Transformer blocks ($J$) | 1, 2 |
| | | Learning rate ($\eta$) | 0.01, 0.001, 0.0001 |
| | | Minibatch size | 64, 128, 256 |
| | | Attention heads ($n_h$) | 1 |
| | | Transformer units ($d_h$) | $1d_{yxa}, 2d_{yxa}, 3d_{yxa}, 4d_{yxa}$ |
| | | Balanced representation size ($d_z$) | $0.5d_{yxa}, 1d_{yxa}, 2d_{yxa}, 3d_{yxa}, 4d_{yxa}$ |
| | | Feed-forward hidden units ($n_{FF}$) | $0.5d_z, 1d_z, 2d_z, 3d_z, 4d_z$ |
| | | Sequential dropout rate ($p$) | 0.1, 0.2 |
| | | Max positional encoding ($l_{max}$) | 15 |
| | | Number of epochs ($n_e$) | 50 |
| RMSNs (Lim et al., 2018) | Propensity treatment network | LSTM layers ($J$) | 1 |
| | | Learning rate ($\eta$) | 0.01, 0.001, 0.0001 |
| | | Minibatch size | 64, 128, 256 |
| | | LSTM hidden units ($d_h$) | $0.5d_{yxa}, 1d_{yxa}, 2d_{yxa}, 3d_{yxa}, 4d_{yxa}$ |
| | | LSTM dropout rate ($p$) | 0.1, 0.2 |
| | | Max gradient norm | 0.5, 1.0, 2.0 |
| | | Number of epochs ($n_e$) | 50 |
| | Propensity history network / Encoder | LSTM layers ($J$) | 1 |
| | | Learning rate ($\eta$) | 0.01, 0.001, 0.0001 |
| | | Minibatch size | 64, 128, 256 |
| | | LSTM hidden units ($d_h$) | $0.5d_{yxa}, 1d_{yxa}, 2d_{yxa}, 3d_{yxa}, 4d_{yxa}$ |
| | | LSTM dropout rate ($p$) | 0.1, 0.2 |
| | | Max gradient norm | 0.5, 1.0, 2.0 |
| | | Number of epochs ($n_e$) | 50 |
| | Decoder | LSTM layers ($J$) | 1 |
| | | Learning rate ($\eta$) | 0.01, 0.001, 0.0001 |
| | | Minibatch size | 256, 512, 1024 |
| | | LSTM hidden units ($d_h$) | $1d_{yxa}, 2d_{yxa}, 4d_{yxa}, 8d_{yxa}, 16d_{yxa}$ |
| | | LSTM dropout rate ($p$) | 0.1, 0.2 |
| | | Max gradient norm | 0.5, 1.0, 2.0, 4.0 |
| | | Number of epochs ($n_e$) | 50 |
| G-Net (Li et al., 2021) | (end-to-end) | LSTM layers ($J$) | 1 |
| | | Learning rate ($\eta$) | 0.01, 0.001, 0.0001 |
| | | Minibatch size | 64, 128, 256 |
| | | LSTM hidden units ($d_h$) | $0.5d_{yxa}, 1d_{yxa}, 2d_{yxa}, 3d_{yxa}, 4d_{yxa}$ |
| | | LSTM output size ($d_z$) | $0.5d_{yxa}, 1d_{yxa}, 2d_{yxa}, 3d_{yxa}, 4d_{yxa}$ |
| | | Feed-forward hidden units ($n_{FF}$) | $0.5d_z, 1d_z, 2d_z, 3d_z, 4d_z$ |
| | | LSTM dropout rate ($p$) | 0.1, 0.2 |
| | | Number of epochs ($n_e$) | 50 |
| GT (ours) | (end-to-end) | Transformer blocks ($J$) | 1, 2 |
| | | Learning rate ($\eta$) | 0.01, 0.001, 0.0001 |
| | | Minibatch size | 64, 128, 256 |
| | | Attention heads ($n_h$) | 1 |
| | | Transformer units ($d_h$) | $1d_{yxa}, 2d_{yxa}, 3d_{yxa}, 4d_{yxa}$ |
| | | Hidden representation size ($d_z$) | $0.5d_{yxa}, 1d_{yxa}, 2d_{yxa}, 3d_{yxa}, 4d_{yxa}$ |
| | | Feed-forward hidden units ($n_{FF}$) | $0.5d_z, 1d_z, 2d_z, 3d_z, 4d_z$ |
| | | Sequential dropout rate ($p$) | 0.1, 0.2 |
| | | Max positional encoding ($l_{max}$) | 15 |
| | | Number of epochs ($n_e$) | 50 |

Table 6: Hyperparameter tuning for all methods on fully synthetic tumor growth data. Here, $d_{yxa} = d_y + d_x + d_a$ is the overall input size. Further, $d_z$ denotes the hidden representation size of our GT, the balanced representation size of CRN (Bica et al., 2020), TE-CDE (Seedat et al., 2022) and CT (Melnychuk et al., 2022), and the LSTM (Hochreiter & Schmidhuber, 1997) output size of G-Net (Li et al., 2021). The hyperparameter grid follows (Melnychuk et al., 2022). Importantly, the tuning ranges for the different methods are comparable. Hence, the comparison of the methods in Section 5 is fair.

## L.2 Hyperparameter tuning: Semi-synthetic data

| Method | Component | Hyperparameter | Tuning range |
|---|---|---|---|
| CRN (Bica et al., 2020) | Encoder | LSTM layers ($J$) | 1,2 |
| | | Learning rate ($\eta$) | 0.01, 0.001, 0.0001 |
| | | Minibatch size | 64, 128, 256 |
| | | LSTM hidden units ($d_h$) | $0.5d_{yxa}, 1d_{yxa}, 2d_{yxa}$ |
| | | Balanced representation size ($d_z$) | $0.5d_{yxa}, 1d_{yxa}, 2d_{yxa}$, |
| | | FF hidden units ($n_{FF}$) | $0.5d_z, 1d_z, 2d_z$ |
| | | LSTM dropout rate ($p$) | 0.1, 0.2 |
| | | Number of epochs ($n_e$) | 100 |
| | Decoder | LSTM layers ($J$) | 1,2 |
| | | Learning rate ($\eta$) | 0.01, 0.001, 0.0001 |
| | | Minibatch size | 256, 512, 1024 |
| | | LSTM hidden units ($d_h$) | Balanced representation size of encoder |
| | | Balanced representation size ($d_z$) | $0.5d_{yxa}, 1d_{yxa}, 2d_{yxa}$ |
| | | FC hidden units ($n_{FF}$) | $0.5d_z, 1d_z, 2d_z$ |
| | | LSTM dropout rate ($p$) | 0.1, 0.2 |
| | | Number of epochs ($n_e$) | 100 |
| TE-CDE (Seedat et al., 2022) | Encoder | Neural CDE hidden layers ($J$) | 1 |
| | | Learning rate ($\eta$) | 0.01, 0.001, 0.0001 |
| | | Minibatch size | 64, 128, 256 |
| | | LSTM hidden units ($d_h$) | $0.5d_{yxa}, 1d_{yxa}, 2d_{yxa}$ |
| | | Balanced representation size ($d_z$) | $0.5d_{yxa}, 1d_{yxa}, 2d_{yxa}$ |
| | | Feed-forward hidden units ($n_{FF}$) | $0.5d_z, 1d_z, 2d_z$ |
| | | Dropout rate ($p$) | 0.1, 0.2 |
| | | Number of epochs ($n_e$) | 100 |
| | Decoder | Neural CDE hidden layers ($J$) | 1 |
| | | Learning rate ($\eta$) | 0.01, 0.001, 0.0001 |
| | | Minibatch size | 256, 512, 1024 |
| | | LSTM hidden units ($d_h$) | Balanced representation size of encoder |
| | | Balanced representation size ($d_z$) | $0.5d_{yxa}, 1d_{yxa}, 2d_{yxa}$ |
| | | Feed-forward hidden units ($n_{FF}$) | $0.5d_z, 1d_z, 2d_z$ |
| | | LSTM dropout rate ($p$) | 0.1, 0.2 |
| | | Number of epochs ($n_e$) | 100 |
| CT (Melnychuk et al., 2022) | (end-to-end) | Transformer blocks ($J$) | 1,2 |
| | | Learning rate ($\eta$) | 0.01, 0.001, 0.0001 |
| | | Minibatch size | 32, 64 |
| | | Attention heads ($n_h$) | 2,3 |
| | | Transformer units ($d_h$) | $1d_{yxa}, 2d_{yxa}$ |
| | | Balanced representation size ($d_z$) | $0.5d_{yxa}, 1d_{yxa}, 2d_{yxa}$ |
| | | Feed-forward hidden units ($n_{FF}$) | $0.5d_z, 1d_z, 2d_z$ |
| | | Sequential dropout rate ($p$) | 0.1, 0.2 |
| | | Max positional encoding ($l_{max}$) | 30 |
| | | Number of epochs ($n_e$) | 100 |
| RMSNs (Lim et al., 2018) | Propensity treatment network | LSTM layers ($J$) | 1,2 |
| | | Learning rate ($\eta$) | 0.01, 0.001, 0.0001 |
| | | Minibatch size | 64, 128, 256 |
| | | LSTM hidden units ($d_h$) | $0.5d_{yxa}, 1d_{yxa}, 2d_{yxa}$ |
| | | LSTM dropout rate ($p$) | 0.1, 0.2 |
| | | Max gradient norm | 0.5, 1.0, 2.0 |
| | | Number of epochs ($n_e$) | 100 |
| | Propensity history network / Encoder | LSTM layers ($J$) | 1 |
| | | Learning rate ($\eta$) | 0.01, 0.001, 0.0001 |
| | | Minibatch size | 64, 128, 256 |
| | | LSTM hidden units ($d_h$) | $0.5d_{yxa}, 1d_{yxa}, 2d_{yxa}$ |
| | | LSTM dropout rate ($p$) | 0.1, 0.2 |
| | | Max gradient norm | 0.5, 1.0, 2.0 |
| | | Number of epochs ($n_e$) | 100 |
| | Decoder | LSTM layers ($J$) | 1 |
| | | Learning rate ($\eta$) | 0.01, 0.001, 0.0001 |
| | | Minibatch size | 256, 512, 1024 |
| | | LSTM hidden units ($d_h$) | $1d_{yxa}, 2d_{yxa}, 4d_{yxa}$ |
| | | LSTM dropout rate ($p$) | 0.1, 0.2 |
| | | Max gradient norm | 0.5, 1.0, 2.0, 4.0 |
| | | Number of epochs ($n_e$) | 100 |
| G-Net (Li et al., 2021) | (end-to-end) | LSTM layers ($J$) | 1,2 |
| | | Learning rate ($\eta$) | 0.01, 0.001, 0.0001 |
| | | Minibatch size | 64, 128, 256 |
| | | LSTM hidden units ($d_h$) | $0.5d_{yxa}, 1d_{yxa}, 2d_{yxa}$ |
| | | LSTM output size ($d_z$) | $0.5d_{yxa}, 1d_{yxa}, 2d_{yxa}$ |
| | | Feed-forward hidden units ($n_{FF}$) | $0.5d_z, 1d_z, 2d_z$ |
| | | LSTM dropout rate ($p$) | 0.1, 0.2 |
| | | Number of epochs ($n_e$) | 100 |
| GT (ours) | (end-to-end) | Transformer blocks ($J$) | 1 |
| | | Learning rate ($\eta$) | 0.001, 0.0001 |
| | | Minibatch size | 32, 64 |
| | | Attention heads ($n_h$) | 2,3 |
| | | Transformer units ($d_h$) | $1d_{yxa}, 2d_{yxa}$ |
| | | Balanced representation size ($d_z$) | $0.5d_{yxa}, 1d_{yxa}, 2d_{yxa}$ |
| | | Feed-forward hidden units ($n_{FF}$) | $0.5d_z, 1d_z, 2d_z$ |
| | | Sequential dropout rate ($p$) | 0.1, 0.2 |
| | | Max positional encoding ($l_{max}$) | 30 |
| | | Number of epochs ($n_e$) | 100 |

Table 7: Hyperparameter tuning for all methods on semi-synthetic data. Here, $d_{yxa} = d_y + d_x + d_a$ is the overall input size. Further, $d_z$ denotes the hidden representation size of our GT, the balanced representation size of CRN (Bica et al., 2020), TE-CDE (Seedat et al., 2022) and CT (Melnychuk et al., 2022), and the LSTM (Hochreiter & Schmidhuber, 1997) output size of G-Net (Li et al., 2021). The hyperparameter grid follows (Melnychuk et al., 2022). Importantly, the tuning ranges for the different methods are comparable. Hence, the comparison of the methods in Section 5 is fair.

