# OpenReview forum: "G-Transformer for Conditional Average Potential Outcome Estimation over Time"
_ICLR.cc/2025/Conference — Submitted to ICLR 2025_

### Official Review · Reviewer_5AUu · 2024-10-30

**Soundness:** 2
**Presentation:** 2
**Contribution:** 2
**Rating:** 5
**Confidence:** 4

**Summary:**

The paper proposes the G-transformer (GT) approach, which combines the G-computation formula and a multi-input transformer to estimate conditional average potential outcomes (CAPO) over time, given time-varying confounding and treatment observational data. GT iteratively generates pseudo-outcomes by reformulating the G-computation formula as recursive conditional expectations. Experimental results on synthetic and semi-synthetic datasets demonstrate that the proposed approach outperforms the baselines in terms of average root mean squared error (RMSE).

**Strengths:**

- The paper addresses the important and impactful real-world problem of estimating CAPO over time under time-varying confounding and treatment from observational data.
- The proposed GT, iteratively generates pseudo-outcomes by reformulating the G-computation formula as recursive conditional expectations, which seems interesting.
- Experimental results on synthetic and semi-synthetic datasets demonstrate that the proposed approach outperforms the baselines in terms of RMSE, especially under conditions of high time-varying confounding and longer CAPO estimation horizons.
- The code and data are publicly available, making it easy to reproduce the results.

**Weaknesses:**

- The novelty is quite limited; the proposed approach could be considered a simple extension of the causal multi-input transformer (Melnychuk et al., 2022) with pseudo-outcome predictions.

- While the paper claims to be the first to use pseudo-outcomes for CAPO estimation over time, several works have already explored the use of pseudo-observations in causal inference and survival analysis, including [1,2,3]. I encourage the author(s) to discuss these works and highlight key differences with their proposed approach.

- The paper asserts that related works, such as G-Net, which properly adjust for time-varying confounding, result in high estimation variance without providing substantial experimental or theoretical results to support this claim. Only RMSE values with error bars are provided (Tables 2 and 3), which seems insufficient given that this is a central claim of the GT approach.

- The experimental results seem underwhelming. Only one semi-synthetic dataset is considered, and RMSE is provided to quantify performance. I encourage the author(s) to explore evaluations tied to meaningful clinical measures, including precision in the estimation of heterogeneous effects, average treatment effects, and qualitative analysis measures of over- or underestimation of outcomes.

- Clarity: The notation and writing need improvement for clarity and readability. The use of both $U$ and $H$ to denote the random variables $\{Y_t, X_t, A_t\}$ does not seem necessary. The paper seems to refer to $H$ throughout, and it's unclear why $U$ was introduced. The definitions for $\bar{Y}$, $\bar{X}$, and $\bar{A}$ are not provided. The recursion could be made more explicit by combining Equations (8) and (9) and providing a simple example, e.g., $\tau = 3$. The paper's use of $\delta$, $\tau$, and $t$ for indexing time needs further simplification. Additionally, the definition for $\tau$ should be provided in the *Setup* section.


**References**
- [1] Andersen et al. (2017), "Causal inference in survival analysis using pseudo-observations", Statistics in medicine.
- [2] Andersen and Perme (2010), "Pseudo-observations in survival analysis", Statistical methods in medical research.
- [3] Chien-Lin et al. (2022), "Causal inference for recurrent event data using pseudo-observations", Biostatistics.

**Minor**
- Table 1: I encourage the author(s) to include citations to related works for improved readability.
- Notation overload:  $\phi$  is used to parameterize both functions $g(\cdot  )$ and $z( \cdot)$.
- Typos: Line 140 should be  $U_{t+\tau}$ remove the 'minus'.
- I encourage the author(s) to increase the font size for tables and figures to be consistent with the main text.
- The definition for $\delta$ in Line 244 is not consistent with that in Line 267.

**Questions:**

- It seems that the GT approach has a clear advantage over existing approaches when $\tau \ge 2$. Could you provide scenarios where estimating CAPO over time ($\tau \ge 2$) has clear clinical benefits?
- Could you quantify the variance of the baseline approaches, for example, using the coefficient of variation?
- Could you provide additional results on the precision in the estimation of heterogeneous effects, average treatment effects, and qualitative analysis measures of over- or underestimation of outcomes?

---

> ### Author Response · Authors · 2024-11-21
>
> Thank you for your helpful feedback on our paper! We are happy that you liked the presentation and the writing of our work. To improve our paper, we **updated our PDF** and highlighted all key changes in **blue color**.
>
> ** **
>
> ### Response to Weaknesses:
>
> ** **
>
> **W1. Novelty of our approach**
>
> Thank you for your comment! It is correct that we leverage a neural backbone that is directly inspired by [4]. This is deliberate, it is a state-of-the–art neural backbone and is designed for medical data sets. Therefore, we think that building on [4] is a strength of our work, as it leverages existing literature.
>
> However, our contribution does **not** lie in our application of transformers. Instead, we present a **novel, end-to-end training algorithm** that leverages iterative G-computation for estimating the conditional average potential outcomes in the time-varying setting. For this, we present a non-trivial generation-learning procedure that correctly targets the CAPO under time-varying confounding (which is different from works such as [4,5,6]). **In particular, Melynchuk et al. **[1]** provide a method that is *biased* as it targets an incorrect estimand, whereas ours targets the correct estimand via adjustments.**
>
> To further stress the importance and novelty of our learning algorithm, we emphasize that we can even instantiate our training algorithm with an LSTM backbone, and we still outperform all baselines clearly (as we show in our **Supplement G.1**). This confirms that our contribution is not the transformer architecture but a novel end-to-end learning algorithm.
>
>
> Below, we offer an in-depth comparison to [4] and thereby spell out clearly how our paper is novel:
>
> [4] does **not** adjust for time-varying confounding. Instead, it only performs back door adjustments, which are insufficient. That is,
>
> $$\mathbb{E}[Y_{t+\tau} [a_{t:t+\tau-1}] | H_t=h_t] \neq \mathbb{E}[Y_{t+\tau} | H_t=h_t, A_{t:t+\tau-1}=a_{t:t+\tau-1}].$$
>
> [4] only tries to adjust for time-varying confounding via balanced representations in an adversarial training objective. This, however, is designed for variance reduction [7] and by no means guarantees that the correct estimand is targeted.
>
> Therefore, it would be irresponsible to deploy [4] in medical practice. In particular, [4] would always be biased, irrespective of the available training data, as it targets an incorrect estimand. Therefore, medical decisions based on the predictions of [4] could be harmful, and at the very least are always biased.
>
> As a remedy, we offer an end-to-end learning procedure that improves over existing literature in that (i) it properly adjusts for time-varying confounding, and (ii) it neither relies on large-variance IPW pseudo-outcomes as in [8], nor on estimating high-dimensional probability distributions as in [9]. In particular, iterative regression-based G-computation has not been explored for estimating CAPOs over time.
>
> **Action:** We added a new clarification where we compare our work against Melnychuk et al and thereby spell out how our work is different and thus novel (see **our revised Section 4.4**).

---

> ### Author Response · Authors · 2024-11-21
>
> **W2. Related work: survival analysis**
>
> Thank you for highlighting these works on survival analysis and causal inference. We are happy to discuss key differences to our work and in particular, why they are **not** applicable to our setting:
>
> There are three important difference between our work and [1]:
>
> **(i)** [1] is aimed at **survival analysis**. In particular, the outcome $Y$ may be the time to an event for a **single, static treatment**. In contrast, our work focuses on estimating the conditional average potential outcome for a sequence of treatments. That is, we are interested in the potential outcome when applying **several treatments** in the future. Both are different estimands and thus require different learning objectives.
>
> **(ii)** [1] is primarily aimed at the **average** causal effect. In contrast, we focus on the **conditional average** effect of a treatment, which is a more complex estimation task. Both are different estimands and thus require different learning objectives.
>
> **(iii)** [1] only investigates *linear* predictors, which is a strong limitation. In contrast, we can capture any *non-linear* DGP thanks to our neural backbone.
>
> There are also important differences between our work and [2]:
>
> **(i)** [2] is also aimed at **survival analysis**, whereas our work focuses on estimating **CAPOs** over time. Both are different estimands and thus require different learning objectives. Importantly, [2] does **not** address time-varying confounding for sequences of treatments and is therefore **not** applicable to our setting.
>
> **(ii)** [2] only considers linear estimators. We deliberately refrained from benchmarking our method with linear estimators as previous work [4] has already shown clear outperformance of neural networks over linear models for estimating CAPOs over time.
>
> Finally, our work differs from [3] in the following ways:
>
> **(i)** [3] focuses on estimating **average causal effects**. In contrast, our work focuses on the more complex task of estimating **conditional average potential outcomes**, which is a more challenging estimation task. Both are different estimands and thus require different learning objectives. Using an estimator for average causal effects in our setting would lead to **severe bias**. In particular, such estimators are completely inapplicable in medical settings as they ignore individual patient characteristics.
>
> **(ii)** [3] only considers estimators such as logistic regressions, GLMs, and ensemble methods. Importantly, they do not leverage the superior function approximation quality of neural networks. Previous works [4] have already shown that linear predictors in particular have severe limitations compared to neural approaches for estimating potential outcomes.
>
> Due to the reasons above, **none** of [1], [2], and [3] are applicable to the setting that we consider in our work.
>
> **Action:** We updated our related work section and discussed pseudo-outcome regression approaches for different purposes than ours. Therein, we included [1], [2], and [3] to our new extended related work in our **new Supplement A** and highlight why these solve an entirely **different task**.

---

> ### Author Response · Authors · 2024-11-21
>
> **W3. Variance**
>
> Thank you for your comment.
>
> Upon reading it, we realized that we should have been more clear about what we mean by low estimation variance. In particular, we were referring to variance relative to the baselines that perform proper adjustments for time-varying confounders:
>
> (i) We thus added a **new Proposition 3** in our paper, which shows that pseudo-outcomes generated by IPW (as in RMSN [8]) have a larger variance than pseudo-outcomes generated by G-computation. We provide a **mathematical proof** for this in **Supplement E**.
>
>
> (ii) We spelled out more clearly the differences between the estimation procedure of G-Net [9] and our GT. That is, our GT is purely **regression based**, whereas G-Net requires estimating the **entire distribution of all time-varying confounders at all time-steps** in the future. We have thus revised our paper by spelling out the limitations of G-Net more clearly.
>
> Formally, for a $\tau$-step-ahead prediction, a $d_y$ dimensional outcome and $d_x$ dimensional covariates, G-Net needs to estimate the (a) the final conditional mean of a $d_y$ dimensional random variable and, additionally, (b) the **full distribution** (that is, **all conditional moments**) of a $(\tau-1)\times d_x \times d_y$ dimensional random variable in order to adjust for confounding. On top, G-Net requires Monte Carlo sampling with so-called hold-out residuals, which further increases variance.
>
> In contrast, our GT only requires performing $(\tau \times d_y$ regressions (that is, only the first conditional moment) to adjust for confounding. Clearly, this estimation task is much simpler and reduces estimation variance considerably. We add details on this difference in our **revised Section 4.4** and highlight the methodological difference in **Supplement F**.
>
> In sum, we improved our work and stated more nuanced claims. In particular, we refrain from making too general claims such as low variance. Instead, we discuss more carefully the limitations of existing methods along three dimensions: existing methods (i) perform simple backdoor adjustments and are thus biased [4,5,6], (ii) rely on IPW and have thus a larger variance pseudo-outcomes than G-computation [8], or (iii) require estimating high-dimensional probability distributions but which is impractical [9].  We are confident that this spells out the limitations of existing baselines in a more precise and clear manner.
>
>
> **Action:** We **changed the wording** throughout our paper. Therein, we removed the general “low variance” statement which may have been perceived as too general and now state the differences along dimensions (i)--(iii) from above. Further, we provide a **new Proposition 3** in our **revised Section 4** that shows that pseudo-outcomes generated through G-computation have lower variance than those generated by IPW. Further, we clarify how the estimation procedure and estimation variance of our GT differs from G-Net in our **revised Section 4**. Finally, we now also benchmark the estimation variance and, for this, report the coefficient of variation in our **new Supplement I**.
>
>
> ** **
>
> **W4. Experimental results:**
>
> Thank you for the suggestions. Based on your feedback, we improved our paper in the following ways:
>
>
>
> * We **performed new experiments for a real-world dataset** from intensive care units using MIMIC-III [10]. We again find that our method has superior performance over all prediction windows.
>
> * We followed your suggestion and **added a new performance metric**, namely, the coefficient of variation. Again, our method performs best.
>
> Our experiments are thus well-aligned with prior literature on CAPO estimation from time series settings in terms of both experiments and performance metrics [4,5,6,8,9].
>
> Below, we provide a detailed answer for each improvement.

---

> ### Author Response · Authors · 2024-11-21
>
> **a. Real-world data:** Thank you. We **added a new experiment on real-world data**. For this, we use the **MIMIC-III dataset** [10], which gives measurements from intensive care units aggregated at hourly levels. Here, we predict the effect of vasopressors and mechanical ventilation on diastolic blood pressure. We find that our **method has superior performance for all prediction windows**. This demonstrates the following: (i) Our method is directly applicable to predict real-world patient outcomes. (ii) Our end-to-end training algorithm does **not** deteriorate performance on factual prediction tasks, that is, without time-varying confounding. This further shows the effectiveness of our approach to G-computation.
>
> **Action:** We **added new experiments with real-world data** based on the MIMIC-III dataset with patient data from intensive care units (see our **new Supplement H).**
>
> ** **
>
> **b. Evaluation metrics:** Thank you! We followed your suggestion and added the coefficient variation. We find that our method performs best. We also emphasize that we followed best practice in evaluating the CAPO in time series settings and thereby closely followed prior literature [4,5,6,8,9].
>
> The choice of the evaluation metric depends on the underlying setting. (i) For example, the precision in the estimation of heterogeneous effects (PEHE) is used for estimating treatment effects, that is, the difference of potential outcomes for two different treatments. (ii) Quantities such as the ATE and APO are designed for at the population level, while we focus on estimates of CAPO (at the individual level). Hence, the average treatment effect would fail to measure the granularity as well as the heterogeneity of our task. Below, we give a more technical explanation of why some of the metrics are not applicable to our setting.
>
> **(i) Precision in estimating heterogeneous effects:** The precision in estimating heterogeneous effects is **not directly relevant** to our task as it refers to the treatment effect and **not** the potential outcome. Estimating conditional average potential outcomes (CAPOs) is a different task than estimating (conditional) average treatment effects (CATEs).
>
> In particular, all methods [4,5,6,8,9] that we benchmark our GT with have the same purpose: they are **all** designed for estimating **CAPOs** over time. Importantly, we are **not** aware of any neural method that is primarily inherently designed for estimating the CATE over time.
>
> Therefore, since our GT performs best among all SOTA baselines for estimating the CAPO $\mathbb{E}[Y_{t+\tau} [a_{t:t+\tau-1}^1] | H_t=h_t]$ for **any** assigned treatment $a_{t:t+\tau-1}^1$, it immediately follows that it **also performs best at estimating the difference of two potential outcomes**
>
> $$\mathbb{E}[Y_{t+\tau} [a_{t:t+\tau-1}^1] | H_t=h_t] - \mathbb{E}[Y_{t+\tau} [a_{t:t+\tau-1}^0] | H_t=h_t],$$
>
> for some baseline treatment $a_{t:t+\tau-1}^0$. Therefore, the PEHE is a **redundant** measure of accuracy compared to our RMSE of the true potential outcome. Upon reading your question, we realized that we should add an explanation on our choice for the RMSE as evaluation metric. We thus provide more clarification on why the RMSE on an individual level is the appropriate metric for evaluating our GT and the baselines in **our revised Section 5**.
>
> **(ii) Average treatment effects (ATEs) / average potential outcomes (APOs):** Estimating ATEs and APOs is a literature stream that may sound similar at first glance but that is very **different** to ours. First, we acknowledge in our related work section that estimating ATEs and APOs is relevant for fields such as epidemiology and economics, and the literature dates back to works such as [11]. Further, there are several methods that are carefully designed to estimate average causal effects [12]. This is **not** the stream of literature we contribute to, and **not** what our GT is designed for.
>
> Second, average causal effects are **not** relevant for **personalized medicine** [13]. In particular, average causal effects **ignore** important patient characteristics such as health conditions, sex, age, social background, etc. Hence, methods for estimating average potential outcomes treat all individuals the same. Personalized medicine is the domain that our GT is designed for and the application domain we seek to contribute to.
>
> Here, the crucial difference is that we need to account for differences in patient characteristics. Therefore, the error in estimating potential outcomes on an **individual level** is the appropriate error metric, and **not** on an average level.

---

> ### Author Response · Authors · 2024-11-21
>
> **(iii) Other measures:** Adding more metrics is a great idea! Therefore, we followed your question Q2 (please see below) and added the **coefficient of variation** as an additional metric. We are not aware of other metrics for measuring over- and underestimation. Should you have a specific suggestion, we are happy to implement them.
>
> **Action:** We clarified our choice for the individual-level RMSE as our primary evaluation metric in our **revised Section 5**. Therein, we reiterated our motivation to estimate the potential outcome on an individual level, as we are interested in personalized medicine. Further, we added the **coefficient of variation** to our previous experimental results in a **new Supplement I.**
>
> ** **
>
> **W5. Clarity:**
>
> Thank you, we followed your suggestion and clarified the notation in our work. In our setup, $U_t$ is meant to be a placeholder for either $A_t$, $X_t$ or $Y_t$. Hence, we wrote $U_t\in\{A_t,X_t,Y_t\}$ such that we do not have to re-specify notation (such as $\bar{X},\bar{Y},\bar{A}$) for each variable separately. That means, $U_t$ is either one of the three variables and additional notation holds for all three of them. In contrast, $H_t$ is referred to as the tuple $H_t = (X_t,Y_t,A_t)$, i.e. the collection of all random variables.
>
> Unfortunately, the definition for $t$, $\delta$ and $\tau$ cannot be further simplified: $t$ refers to the time when our intervention starts, $\tau$ is the prediction horizon, and $\delta$ refers to a running index to any point in time in between, which we need for our iterative G-computation.
>
> **Action:** We clarified the notation around $U_t$ in our **revised Section 3**. We further introduced $\tau$ explicitly. Finally, we highlight our step-by-step examples in **Supplement D.3 **in our **revised Section 4**.
>
> **Minor:**
>
> Thank you for your suggestions!
>
> **Actions:**
>
>
>
> * We included citations in our tables.
> * We used $\phi$ and $\theta$ for the different weights of our network.
> * The “minus” was due to an “overline” of the line below. We fixed the display problem.
> * We increased the font sizes of the tables.
> * We fixed L. 267, thank you!

---

> ### Author Response · Authors · 2024-11-21
>
> ### Response to Questions:
>
> **Q1. $\tau\geq 2$ step ahead prediction:**
>
> Thank you for your question. Our method is carefully designed to successfully help answer counterfactual questions like “What is the outcome of patient X if she received treatments A, B, and C over the next 5 days, given her personalized information?” Importantly, in medicine, treatments are almost always applied **sequentially** [14,15]. As observational data becomes more and more available [13,16], there is growing interest in estimating the effect of treatments from EHRs [13,16,17] or wearable devices [18]. Hence, we need methods that can estimate the effect of treatment sequences over time and properly adjust for time-varying confounding.
>
>
> Methods for estimating CAPOs in the static setting are very restrictive for personalized medicine in that they ignore any time-varying component. In particular, they do **not** account for time-varying confounding that arises when estimating the CAPO for **sequences of treatments**. Time-varying confounding means that past treatments influence future covariates, which, in turn, affect both subsequent treatment assignment and outcomes. This feedback loop complicates the estimation of conditional average potential outcomes.
>
> Our GT adjusts for this confounding bias and is, thus, of direct practical relevance for personalized medicine.
>
> **Action:** We added a **new Supplement B**, where we discuss the relevance of our scenario and how our method could be successfully incorporated into medical practice.
>
> **Q2. Coefficient of variation:**
>
> Thank you for this suggestion!
>
> **Action:** We think this is a great idea and therefore added the coefficient of variation to our previous studies (see **our new Supplement I**).
>
> **Q3. Additional metrics:**
>
> Thank you for your question. We followed best practice in evaluating the CAPO in time series settings and thereby closely followed prior literature [4,5,6,8,9]. In particular, the CAPO is the metric that our GT and all baselines are designed to estimate. Further, the choice of the metric depends on the underlying setting. (i) Specifically, the precision in the estimation of heterogeneous effects (PEHE) is widely used for estimating treatment effects, and is a **redundant** metric in our setting. (ii) Average treatment effects and average potential outcomes are designed at the **population** level, while we focus on estimates of CAPO, that is, at the **individual** level. Hence, the average level metrics would **fail** to measure the granularity as well as the heterogeneity of our task. (iii) The idea of over- and underestimation is intriguing but we are not aware of corresponding metrics in the literature. Should you have a specific suggestion, we are happy to implement them.
>
> We kindly refer to our response in **W4.** Upon reading your question, we realized that we should add an explanation where we describe in detail the motivation for our evaluation metric.
>
> **Action:** We provide more clarification on why the RMSE on an individual level is the appropriate metric for evaluating our GT and the baselines in our **revised Section 5.**

---

> ### Author Response · Authors · 2024-11-21
>
> [1] Andersen et al. (2017), "Causal inference in survival analysis using pseudo-observations", Statistics in medicine.
>
> [2] Andersen and Perme (2010), "Pseudo-observations in survival analysis", Statistical methods in medical research.
>
> [3] Chien-Lin et al. (2022), "Causal inference for recurrent event data using pseudo-observations", Biostatistics.
>
> [4] Melnychuk, Valentyn, Dennis Frauen, and Stefan Feuerriegel. "Causal transformer for estimating counterfactual outcomes." International Conference on Machine Learning. PMLR, 2022.
>
> [5] Ioana Bica, Ahmed M. Alaa, James Jordon, and Mihaela van der Schaar. Estimating counterfactual treatment outcomes over time through adversarially balanced representations. In ICLR, 2020.
>
> [6] Toon Vanderschueren, Alicia Curth, Wouter Verbeke, and Mihaela van der Schaar. Accounting for informative sampling when learning to forecast treatment outcomes over time. In ICML, 2023.
>
> [7] Fredrik Johansson, Uri Shalit, and David Sontag. Learning representations for counterfactual inference. In ICML, 2016.
>
> [8] Bryan Lim, Ahmed M. Alaa, and Mihaela van der Schaar. Forecasting treatment responses over time using recurrent marginal structural networks. In NeurIPS, 2018.
>
> [9] Rui Li, Stephanie Hu, Mingyu Lu, Yuria Utsumi, Prithwish Chakraborty, Daby M. Sow, Piyush Madan, Jun Li, Mohamed Ghalwash, Zach Shahn, and Li-wei Lehman. G-Net: A recurrent network approach to G-computation for counterfactual prediction under a dynamic treatment regime. In ML4H, 2021.
>
> [10] Alistair E. W. Johnson, Tom J. Pollard, Lu Shen, Li-wei H. Lehman, Mengling Feng, Mohammad Ghassemi, Benjamin Moody, Peter Szolovits, Leo Anthony Celi, and Roger G. Mark. MIMIC-III, a freely accessible critical care database. Scientific Data, 3(1):160035, 2016.
>
> [11] James M. Robins. Correcting for non-compliance in randomized trials using structural nested mean models. Communications in Statistics - Theory and Methods, 23(8):2379–2412, 1994.
>
> [12] Dennis Frauen, Tobias Hatt, Valentyn Melnychuk, and Stefan Feuerriegel. Estimating average causal effects from patient trajectories. In AAAI, 2023.
>
> [13] Stefan Feuerriegel, Dennis Frauen, Valentyn Melnychuk, Jonas Schweisthal, Konstantin Hess, Alicia Curth, Stefan Bauer, Niki Kilbertus, Isaac S. Kohane, and Mihaela van der Schaar. Causal machine learning for predicting treatment outcomes. Nature Medicine, 30:958–968, 2024.
>
> [14] Apperloo, E.M., Gorriz, J.L., Soler, M.J. *et al.* Semaglutide in patients with overweight or obesity and chronic kidney disease without diabetes: a randomized double-blind placebo-controlled clinical trial. Nature Medicine, 2024.
>
> [15] Stefanie Schüpke et al. Ticagrelor or Prasugrel in Patients with Acute Coronary Syndromes. The New England Journal of Medicine, 2019.
>
> [16] Ahmed Allam, Stefan Feuerriegel, Michael Rebhan, and Michael Krauthammer. Analyzing patient trajectories with artificial intelligence. Journal of Medical Internet Research, 23(12):e29812, 2021.
>
> [17] Ioana Bica, Ahmed M. Alaa, Craig Lambert, and Mihaela van der Schaar. From real-world patient data to individualized treatment effects using machine learning: Current and future methods to address underlying challenges. Clinical Pharmacology and Therapeutics, 109(1):87–100, 2021.
>
> [18] Samuel L. Battalio et al. Sense2Stop: A micro-randomized trial using wearable sensors to optimize a just-in-time-adaptive stress management intervention for smoking relapse prevention. Contemporary Clinical Trials, 109:106534, 2021.

---

> > ### Comment · Reviewer_5AUu · 2024-11-26
> > **Response by Reviewer**
> >
> > Thank you for the rebuttal and for providing additional experimental and theoretical results. The changes to the paper seem too substantial compared to the original submission, and I believe the paper would benefit from another round of review. For these reasons, I am maintaining my score.

---

### Official Review · Reviewer_i9bM · 2024-11-01

**Soundness:** 3
**Presentation:** 4
**Contribution:** 1
**Rating:** 3
**Confidence:** 5

**Summary:**

This work proposed a framework to estimate conditional average potential outcome over time using a combination of transformer and iterative pseudo-outcome regression. The framework is tested in experiments on a synthetic dataset based on tumor growth model and a semi-synthetic dataset based on MIMIC-III, and is shown to outperform several benchmark models on these two datasets.

**Strengths:**

1. Paper is well written, problem formulation, architecture descriptions and the presentation of learning procedures are all clearly stated and very easy to follow.

2. The results look compeling. Table 2 and 3 showed noticable improvement of the proposed model over the compared benchmarks.

**Weaknesses:**

1. The novelty of this work is the most important concern to me. The combination of transformer and temporal-difference learning (i.e. pseudo-outcome regression as called by the authors) are already proposed in [1] for the estimation of CAPO (on top of that, [1] also went one step further and proposed the adjustment for marginal causal parameters). Besides, the heterogeneous token transformer architecture in [1] appears to be a more clean and intuitive solution as opposed to the proposed architecture in my judgement. The proposed transformer architecture is rather trivial and does not significantly differ from [2]. At the very least, the author should compare their architeture to [1], show experimental results and properly state their contributions compared to prior work.

2. The paper has a very weak description of prior work related to pseudo-outcome regression in my opinion. Not only is the exact same procedure temporal-difference learning proposed before as stated above, it is widely used in causal inference and reinforcement learning known as sequential regression [3] and Q-learning [4]. The omission of these works by the authors made it look like pseudo-outcome regression is a novel contribution of their own.

[1] Shirakawa, Toru, et al. "Longitudinal Targeted Minimum Loss-based Estimation with Temporal-Difference Heterogeneous Transformer." International Conference on Machine Learning. PMLR, 2024.

[2] Melnychuk, Valentyn, Dennis Frauen, and Stefan Feuerriegel. "Causal transformer for estimating counterfactual outcomes." International Conference on Machine Learning. PMLR, 2022.

[3] van der Laan, M. J. and Gruber, S. Targeted Minimum Loss Based Estimation of Causal Effects of Multiple Time Point Interventions. The International Journal of Biostatistics, 8(1), 2012.

[4] Chebotar, Yevgen, et al. "Q-transformer: Scalable offline reinforcement learning via autoregressive q-functions." Conference on Robot Learning. PMLR, 2023.

**Questions:**

My suggestions correspond to the two weaknesses above. I think comparison to prior work that already proposed the combination of transformer and temporal-difference learning/pseudo-outcome regression is absolutely crucial; literature review related to pseudo-outcome regression ([1][3][4] and other related work) should be added, and contribution of this work should be properly and correctly stated.

---

> ### Author Response · Authors · 2024-11-21
>
> Thank you for your helpful feedback on our paper! We are happy that you liked the presentation and the writing of our work. To improve our paper, we **updated our PDF** and highlighted all key changes in **blue color**.
>
> ** **
>
> ### Response to Weaknesses & Questions:
>
> ** **
>
> **W1. Non-trivial contribution and temporal difference learning**
>
> Thank you for pointing out the work on temporal difference learning [1] and the causal transformer [2]. We are happy to provide more clarity on how our work differs from theirs:
>
>
> **Comparison to [1]:** In short, paper [1] aims at a **different task**: the **average** potential outcome (APO), while our paper aims at the **conditional average** potential outcome (CAPO). The APO predicts the outcome of a treatment as a **population-wide quantity**. In contrast, the CAPO is a different estimand that allows for **fine-grained** predictions of outcome at the individualized level. Hence, **our estimand is more fine-grained, more complex, and is relevant in clinical practice where the aim is to personalize treatment decisions to individual patients**.
>
> Below, we provide a more technical discussion of the methodological differences. While the transformer in [1] and ours are similar in that both use transformers, they are designed for entirely different purposes: The transformer in [1] is inherently designed for efficient estimation of **average potential outcomes** (APOs) and **not** conditional average potential outcomes (**C**APOs) as our GT. Here, [1] is explicitly **biased by sequentially targeting** the APO. In contrast, our work does **not** require any targeting step. On top, [1] requires estimation of **additional nuisance** parameters such as the propensity scores which our method does **not**. Further, [1] is only **evaluated for estimating APOs** and only against baselines for APOs. Hence, [1] is not designed for a completely different purpose and, thus, is biased for our setting. We further do not see how their transformer could be extended for our setting.
>
> Instead, our GT is designed to estimate the expected potential outcome for a future sequence of treatments, **conditionally** on the observed history. This is a more challenging task, which we solve through novel end-to-end training in a single neural architecture. Additionally, we show that our approach is not only applicable to a transformer backbone but **agnostic** to the neural backbone such that it can also be applied to simple LSTM backbones (**Supplement G.1**).
>
> **Action:** We cite [1] and offer a more detailed comparison to show that our task is completely different, and, thereby, we spell out clearly that our method is novel (see **our revised Section 2**).

---

> ### Author Response · Authors · 2024-11-21
>
> **Comparison to [2]:** In short, **paper [2] learns an incorrect objective for our task and thus generates estimates that are biased**. In contrast, we propose a **novel, end-to-end training algorithm** that leverages iterative G-computation for estimating the conditional average potential outcomes in the time-varying setting. For this, we present a non-trivial generation-learning procedure that correctly estimates the CAPO under time-varying confounding (which is different from works such as [2]). In other words, our contribution does **not** lie in our application of transformers, but rather in the end-to-end training algorithm and the end-to-end architecture.
>
> In the following, we provide more detail on the differences. [2] uses transformers for the same purpose as our work, and, hence, it is **one of our baselines**. However, [2] performs a simple backdoor adjustment for estimating the CAPO over time, which is **insufficient and leads to estimates that are always biased** [4]. Formally, we have
>
> $$\mathbb{E}[Y_{t+\tau} [a_{t:t+\tau-1}] | H_t=h_t] \neq \mathbb{E}[Y_{t+\tau} | H_t=h_t, A_{t:t+\tau-1}=a_{t:t+\tau-1}].$$
>
> The authors of [2] only employ balancing in an adversarial training objective. This, however, is designed for variance reduction [3] and is **not** an adjustment for time-varying confounding. In other words, their results are **biased**.
>
> As a remedy, we proposed an end-to-end training algorithm based on a neural generation-learning procedure  for effectively estimating the CAPO for sequences of treatments. To this end, our contribution over [2] is not the architecture but the learning algorithm that is different and novel: **we provide a new and completely different end-to-end training algorithm** based on a novel generation-learning procedure that properly adjusts for time-varying confounding and that targets the correct estimand. Our learning algorithm is flexible: it is also applicable to LSTM backbones and outperforms all baselines clearly (as we show in **Supplement G.1**). This confirms that our contribution is not the transformer architecture but a novel end-to-end learning algorithm.
>
> **Action:** We clarified our contribution over [2] in that we propose an end-to-end training algorithm based on a novel generation-learning procedure. Further, we included [1] in our related work and discussed similarities and important differences. Therein, we discussed why [1] is not designed for our scenario, that is, estimating CAPOs over time. Specifically, we **revised Section 2** where we spell out the differences in greater depth and implications for medicine. In particular, using [1] and [2] in clinical practice would therefore be dangerous, as estimates would always be biased, irrespective of the amount of available data. Further, we discuss the applicability of our method for medical applications in our **new Supplement B**. We are convinced that **proposing a novel, neural end-to-end training algorithm** is relevant for improving treatment decision-making in clinical practice
>
> ** **
>
> **W2. Related work**
>
> Thank you. We are happy to provide more related literature on Q-learning as well as more works on estimating average potential outcomes in the time-varying setting. We did not intend to claim that pseudo-outcome regression is our novelty, and we apologize if we conveyed the impression. Rather, we clarified that our main contribution is proposing a neural training algorithm for estimating CAPOs in the time-varying setting that is directly applicable in medical scenarios.
>
> **Action:** We carefully revised our related work section (see **our revised Section 2**). Therein, we added **additional literature on estimating APOs** via pseudo-outcome regression. On top, we added an extensive **discussion on the similarities and differences between Q-learning and G-computation** in an extended related work section in our **new Supplement A.**
>
>
> ** **
>
> [1] Shirakawa, Toru, et al. "Longitudinal Targeted Minimum Loss-based Estimation with Temporal-Difference Heterogeneous Transformer." International Conference on Machine Learning. PMLR, 2024.
>
> [2] Melnychuk, Valentyn, Dennis Frauen, and Stefan Feuerriegel. "Causal transformer for estimating counterfactual outcomes." International Conference on Machine Learning. PMLR, 2022.
>
> [3] Fredrik Johansson, Uri Shalit, and David Sontag. Learning representations for counterfactual inference. In ICML, 2016.
>
> [4] Dennis Frauen, Konstantin Hess, and Stefan Feuerriegel. Model-agnostic meta-learners for estimating heterogeneous treatment effects over time. arXiv preprint, 2024.

---

> ### Comment · Reviewer_i9bM · 2024-11-25
> **Response to Authors**
>
> Thank the authors for the detailed response, however, I respectfully couldn't disagree more with your opinion. Here is the contributions of [1]:
>
> - Proposed Transformer + Temporal Difference Learning (i.e. Pseudo Outcome Regression) for the estimation of CAPO
>
> - Proposed using the above estimation of CAPO + Targeted Learning for the estimation of APO
>
> Here is the contribution of this work:
>
> - Proposed Transformer + Pseudo Outcome Regression (i.e. Temporal Difference Learning) for the estimation of CAPO
>
> I mean your contribution is literally a subset of [1] and in my judgement, you can't just say because the end goal of [1] is different then you have different contributions -- that's like taking paper [1] and remove its contribution 2 and publish a new paper. To me that doesn't really make sense and I would like to maintain my score for that reason.
>
> [1] Shirakawa, Toru, et al. "Longitudinal Targeted Minimum Loss-based Estimation with Temporal-Difference Heterogeneous Transformer." International Conference on Machine Learning. PMLR, 2024.

---

### Official Review · Reviewer_5WFT · 2024-11-03

**Soundness:** 4
**Presentation:** 4
**Contribution:** 3
**Rating:** 8
**Confidence:** 4

**Summary:**

This study proposes G-Transformer to estimate conditional average potential outcomes (CAPOs) over time while addressing two major limitations of the previous studies: (1) adjustments for time-varying confounders; (2) and large estimation variance. Particularly, the authors propose a viable solution to G-computation, iterative pseudo-outcome regression, using Transformer.

**Strengths:**

The study is motivated and the problem is well-defined by the research gap of the previous studies. Specifically, the authors address the limitations of the previous studies regarding G-computation to estimate CAPOs with neural networks by providing a viable solution of iterative pseudo-outcome regression, although the neural network architecture itself is similar to the existing study (Melnychuk et al., 2022).

The author provides sufficient and proper supplementary materials to explain the details of the study.

**Weaknesses:**

The reviewer did not find any significant weaknesses.

The reviewer thinks a more principled and diverse way to evaluate the quality of the generated pseudo-outcomes will be desirable. While the experiment conducted in E.2 could empirically show the learning step is sensitive to the noise of the generated pseudo-outcomes, it might not be sufficient to show the quality of the generated pseudo-outcomes as it is titled. An ablation study that separates each step (i.e., another step is frozen when one step is trained) might be useful for a controlled experiment to measure the sensitivity of the performance in terms of how much the model is trained to generate the pseudo-outcomes.

**Questions:**

Table 2 and 3 have low readability due to small font sizes.

The reviewer thinks Supplementary E.2 regarding the quality of pseudo-outcome is worth being mentioned in the main text

---

> ### Author Response · Authors · 2024-11-21
>
> Thank you for your positive review of our paper! Thank you for your detailed feedback on our manuscript. We took all your comments at heart and improved our manuscript accordingly. We **updated our PDF** and highlighted all key changes in **blue color**.
>
>
> ### Response to Weaknesses & Questions:
>
> **1. Pseudo outcomes:**
>
> Thank you for your comment. Upon reading your comment, we realized that we could have opted for a better name for Supplement E.2. Hence, we followed your suggestion and renamed Supplement E.2 to “Sensitivity to noise in pseudo-outcomes”.
>
> We agree that it is important to evaluate the quality of the pseudo outcomes. In our current study, we artificially corrupted the pseudo outcomes and noticed a significant decrease in performance. Without corruption, the performance is good, which means, in turn, that the generated pseudo outcomes must be meaningful.
>
>
> Freezing certain parts of our GT during training is an interesting idea! We carefully discussed this idea among our author team. One challenge is that our GT is designed in a way that is trained end-to-end. Hence, freezing the training of one of the G-computation heads essentially means that we are artificially corrupting the quality of the pseudo-outcomes they generate. If a G-computation head is not properly trained, it will produce a poor pseudo outcome.
>
> However, we agree that “Sensitivity to noise in pseudo-outcomes” is a better title for E.2 and thank the reviewer for their suggestion!
>
> **Action:** We changed the name of our Supplement G.2 to “Sensitivity to noise in pseudo-outcomes” and referenced Supplement G.2 in our main paper. Thank you!
>
> ** **
>
> **2. Font size:**
>
> Thank you for noticing this.
>
>
> **Action:** In our updated version, we increased the font sizes of Tables 2 and 3.

---

### Official Review · Reviewer_AaBK · 2024-11-03

**Soundness:** 2
**Presentation:** 3
**Contribution:** 2
**Rating:** 5
**Confidence:** 4

**Summary:**

The paper studies conditional average potential outcome estimation over time. The authors propose a model called G-Transformer for addressing two primary challenges in the field: adjustment for time-varying confounders and low variance in estimates. The authors validate their approach through experiments on synthetic and semi-synthetic data, demonstrating improvements over current baselines.

**Strengths:**

- The paper provides a clear, detailed description of model architecture.

- The authors conduct extensive experiments with synthetic and semi-synthetic datasets, demonstrating GT's effectiveness under different levels of confounding settings.

- The code is provided.

**Weaknesses:**

- The contribution of this paper is somewhat limited.
  * From the methodology perspective, the proposed method's improvement over existing approaches appears marginal. The primary contribution is exploring the transformer backbone in treatment effect estimation, which has been widely investigated in existing work (static setting [1] and dynamic setting [2]).
  * From the application perspective, the paper claims that their model can help personalize decision-making from patient trajectories in medicine. However, it lacks discussion on how the model's estimates would concretely translate to real-world decision support. Furthermore, the assumptions on which the model relies (e.g., positivity, ignorability) are restrictive and may not hold robustly in practical, real-world scenarios.

- While the paper claims to reduce estimation variance, it lacks a theoretical analysis of variance bounds. Given GT’s objective to minimize estimation variance, a more rigorous theoretical justification or a derivation of variance bounds would significantly strengthen this claim.

-  The paper does not consider the existence of unobserved (and potentially time-varying) confounders, which can introduce biases and affect CAPO estimation in real-world EHR settings.

- The study uses a semi-synthetic dataset derived from MIMIC-III. It would be valuable to see the model's performance on fully real-world datasets to assess its utility in practical clinical decision-making problems.

[1] Zhang, Yi-Fan, et al. "Exploring transformer backbones for heterogeneous treatment effect estimation." arXiv preprint arXiv:2202.01336 (2022).

[2] Melnychuk, Valentyn, Dennis Frauen, and Stefan Feuerriegel. "Causal transformer for estimating counterfactual outcomes." International Conference on Machine Learning. PMLR, 2022.

**Questions:**

See weaknesses above.

---

> ### Author Response · Authors · 2024-11-21
>
> Thank you for your helpful review of our paper! Thank you for your detailed feedback on our manuscript. We took all your comments at heart and improved our manuscript accordingly. We **updated our PDF** and highlighted all key changes in **blue color**.
>
> ** **
>
> ### Responses to Weaknesses & Questions
>
>
>
> **1. Contribution**
>
> Thank you for giving us the opportunity to carefully spell out the novelty of our work and where our contributions are.
>
> Our contribution does **not** lie in our application of transformers. Instead, we present a **novel, end-to-end training algorithm** that leverages iterative G-computation for estimating the conditional average potential outcomes in the time-varying setting. For this, we present a non-trivial generation-learning procedure that correctly targets the CAPO under time-varying confounding (which is different from works such as [2,3,11]).
>
> So far, **existing methods for estimating CAPOs over time have key limitations**: either they (i) perform simple backdoor adjustments and are thus biased [2,3,11], (ii) rely on IPW [10] and have thus a larger variance than G-computation (as we show in **Supplement E**), or (ii) require estimating high-dimensional probability distributions but which is ineffective [4] (**Supplement F**). We address all three limitations (i)--(iii) by presenting a regression-based, end-to-end training procedure for G-computation.
>
> Our learning algorithm is flexible: it is also applicable to **LSTM backbones** and outperforms all baselines clearly (as we show in **Supplement G.1**). This confirms that our contribution is not the transformer architecture but a novel end-to-end learning algorithm.
>
> ** **
>
> **1.a Transformer backbone:**
> Thank you for your comment. Although we use a transformer backbone in our method, this is not the primary contribution of our work. In line with this, we explicitly state that our backbone is directly inspired by state-of-the-art transformers such as [2]. In **Supplement G.1**, we also show that our training algorithm is directly applicable to simple LSTM backbones. The results with the LSTM backbone outperform all baselines by a large margin, which confirms that our contribution is not the architecture but a novel end-to-end learning algorithm.
>
> To this end, **our main novelty is our novel, end-to-end learning algorithm: we leverage iterative pseudo-outcome regression** in a **generation-learning procure** for CAPOs over time (see our **new Section 4.4**). We are not aware of any work that has developed a neural approach to G-computation through iterative regressions. So far, only G-Net has explored G-computation for CAPOs. Yet, their approach is entirely different from ours. In particular, G-Net estimates the full distribution of all future, time-varying confounders, which is highly ineffective (**Supplement F**). In contrast, we design a novel training algorithm that is neural and that, for the first time, allows for end-to-end learning.
>
>
>
> **We further emphasize that our work is completely different from [1] and [2]:**
>
> [1] explores transformers for causal inference in the **static setting**. Here, simple backdoor adjustments are sufficient to adjust for confounding. This is **entirely different in the time-varying setting**, as **time-varying confounding** requires more complex adjustments such as G-computation or IPW. Hence, the setting in [1] is  much simpler. In our work, we show how to effectively estimate the G-computation formula in a **novel training algorithm** that leverages iterative generation-learning steps in a **neural end-to-end generation-learning procedure**.
>
> [2] uses transformers for the same purpose as our work, and, hence, it is one of our baselines. However, [2] performs a simple backdoor adjustment for estimating the CAPO over time, which is **insufficient and leads to estimates that are biased** [12]**.** That is, **we can even show that their objective is biased** **as the authors target an incorrect estimand**. Formally, we have
>
> $$\mathbb{E}[Y_{t+\tau} [a_{t:t+\tau-1}] | H_t=h_t] \neq \mathbb{E}[Y_{t+\tau} | H_t=h_t, A_{t:t+\tau-1}=a_{t:t+\tau-1}].$$
>
> The authors in [2] only employ balancing in the form of an adversarial objective, which, however, only reduces finite sample estimation variance [13]. More importantly, this does not, by any means, help targeting the correct estimand and is thus biased. While our backbone is inspired by [2], our work is different and novel: **we provide a new and completely different end-to-end training algorithm** based on a novel generation-learning procedure that properly adjusts for time-varying confounding and that targets the **correct estimand**.
>
> **Action:** We have **reworked our Section 2 and Section 4** to spell out more clearly where existing literature ends and where our novelty starts.

---

> ### Author Response · Authors · 2024-11-21
>
> **1.b Real-world decision support and assumptions:**
> Thank you for pointing this out. Our method is carefully designed to successfully help answer counterfactual questions like “What is the outcome of patient X if she received treatments A, B, and C over the next 5 days, given her personalized information?” Importantly, in medicine, treatments are almost always applied **sequentially** [14,15]. As observational data becomes more and more available [16,17], there is growing interest in estimating the effect of treatments from EHRs [16,17,19] or wearable devices [18]. Hence, we need methods that can estimate the effect of treatment sequences over time and properly adjust for time-varying confounding.
>
> Our model makes the three assumptions (i) consistency, (ii) positivity, and (iii) sequential ignorability. These assumptions are **standard in the literature** [2,3,4,11]. Other works often make even stricter assumptions than ours (e.g., they even impose additional Markov assumptions [20]), while, in contrast, ours are weaker. Further, (i)-(iii) are the time-varying analogue to the standard assumptions in the static setting [5,6]. Importantly, the works that operate in the **static setting** implicitly make much more **unrealistic** assumptions: they impose that treatments are only applied once and that covariates and outcomes do not evolve over time, thereby biasing future treatment decisions. Instead, our time-varying setting is much more realistic.
>
> Finally, we are positive that our assumptions are not too restrictive in medical reality. (i) Consistency is typically guaranteed as long as data is properly recorded. (ii) Positivity can be guaranteed by properly filtering the data or by using propensity clipping. On top,  this assumption becomes less restrictive as the overall size of observational datasets grows. (iii) Finally, there are efforts in sensitivity analysis [7,8] and partial identification [9] to relax the ignorability assumption, but these works are orthogonal to our approach. In practice, our method would be part of a well established causal inference pipeline ((1) point estimation -> (2) uncertainty quantification -> (3) sensitivity analysis). Further, while sequential ignorability is still commonly made in epidemiology [21], many studies in digital health interventions can satisfy this assumption by design. Needless to say, the assumption becomes more plausible with better and more comprehensive data measurement in clinical practice.
>
>
> **Action:** We added a **new Supplement B**, where we discuss how our method could be successfully incorporated into medical practice. We also discuss the plausibility of our assumptions in practice and emphasize that the assumptions are standard in causal inference, and a relaxation of the single-time intervention in the static setting (see **our revised Section 3**).

---

> ### Author Response · Authors · 2024-11-21
>
> **2. Estimation variance:**
>
> Thank you for your comment. Upon reading it, we realized that we should have been more clear about what we mean by low estimation variance. In particular, we were referring to variance relative to the baselines that perform proper adjustments for time-varying confounders:
>
> (i) We thus added a **new Proposition 3** in our paper, which shows that pseudo-outcomes generated by IPW (as in RMSN [10]) have a larger variance than pseudo-outcomes generated by G-computation. We provide a **mathematical proof** for this in **Supplement E**.
>
>
> (ii) We spelled out more clearly the differences between the estimation procedure of G-Net [4] and our GT. That is, our GT is purely **regression based**, whereas G-Net requires estimating the **entire distribution of all time-varying confounders at all time-steps** in the future. We have thus revised our paper by spelling out the limitations of G-Net more clearly.
>
> Formally, for a $\tau$-step-ahead prediction, a $d_y$ dimensional outcome and $d_x$ dimensional covariates, G-Net needs to estimate the (a) the final conditional mean of a $d_y$ dimensional random variable and, additionally, (b) the **full distribution** (that is, **all conditional moments**) of a $(\tau-1)\times d_x \times d_y$ dimensional random variable in order to adjust for confounding. On top, G-Net requires Monte Carlo sampling with so-called hold-out residuals, which further increases variance.
>
> In contrast, our GT only requires performing $(\tau \times d_y)$ regressions (that is, only the first conditional moment) to adjust for confounding. Clearly, this estimation task is much simpler and reduces estimation variance considerably. We add details on this difference in our **revised Section 4** and highlight the methodological difference in **Supplement F**.
>
> In sum, we improved our work and stated more nuanced claims. In particular, we refrain from making too general claims such as low variance. Instead, we discuss more carefully the limitations of existing methods along three dimensions: existing methods (i) perform simple backdoor adjustments and are thus biased, (ii) rely on IPW and have thus a larger variance pseudo-outcomes than G-computation, or (iii) require estimating high-dimensional probability distributions but which is impractical.  We are confident that this spells out the limitations of existing baselines in a more precise and clear manner.
>
>
> **Action:** We **changed the wording** throughout our paper. Therein, we removed the general “low variance” statement which may have been perceived as too general and now state the differences along dimensions (i)--(iii) from above. Further, we provide a **new Proposition 3** in our **revised Section 4** that shows that pseudo-outcomes generated through G-computation have lower variance than those generated by IPW. Finally, we clarify how the estimation procedure and estimation variance of our GT differs from G-Net in our **revised Section 4** and **Supplement F**.
>
>
> ** **
>
> **3. Unobserved confounding:**
>
> Thank you for your comment. This is correct – we do not consider unobserved confounding. Our three assumptions (i) consistency, (ii) positivity, and (iii) sequential ignorability are standard in the literature [2,3,4,11] and the time-varying analogue to the static setting [5,6].
> Further, we are that our assumptions are not too restrictive on medical reality. We kindly refer to our answer in **W1.b**.
>
>
> **Action:** We discuss the plausibility of our assumptions in greater detail and give concrete examples where these are met (see our revised Section 3).  We further added a **new Supplement B** where we discuss practical considerations for using our methods. Therein, we highlight the importance of carefully checking assumptions before implementing our method in clinical practice.
>
> ** **
>
> **4. Real-world data:**
>
> Thank you for this suggestion! We are more than happy to demonstrate our method using a real-world prediction task. Therefore, we **added a new experiment on real-world data**. Thereby, we show that our method performs well for predicting patient outcomes on factual data. For this, we use the **MIMIC-III dataset** [22], which gives measurements from intensive care units aggregated at hourly levels. Here, we predict the effect of vasopressors and mechanical ventilation on diastolic blood pressure. We find our **method has superior performance for all prediction windows**. This demonstrates the following: (i) Our method is directly applicable to predict real-world patient outcomes. (ii) Our end-to-end training algorithm does **not** deteriorate performance on factual prediction tasks, that is, **without** time-varying confounding. This further shows the effectiveness of our approach to G-computation.
>
> **Action:** We **added new experiments with real-world data** based on the MIMIC-III dataset with patient data from intensive care units (see our **new Supplement H in our revised PDF**).

---

> ### Author Response · Authors · 2024-11-21
>
> 1] Yi-Fan Zhang, Hanlin Zhang, Zachary C. Lipton, Li Erran Li, Eric P. Xing. Exploring transformer backbones for heterogeneous treatment effect estimation. arXiv preprint arXiv:2202.01336, 2022.
>
> [2] Valentyn Melnychuk, Dennis Frauen, and Stefan Feuerriegel. Causal transformer for estimating counterfactual outcomes. In ICML, 2022.
>
> [3] Nabeel Seedat, Fergus Imrie, Alexis Bellot, Zhaozhi Qian, and Mihaela van der Schaar. Continuous-time modeling of counterfactual outcomes using neural controlled differential equations. In ICML, 2022.
>
> [4] Rui Li, Stephanie Hu, Mingyu Lu, Yuria Utsumi, Prithwish Chakraborty, Daby M. Sow, Piyush Madan, Jun Li, Mohamed Ghalwash, Zach Shahn, and Li-wei Lehman. G-Net: A recurrent network approach to G-computation for counterfactual prediction under a dynamic treatment regime. In ML4H, 2021.
>
> [5] Krikamol Muandet, Montonobu Kanagawa, Sorawit Saengkyongam, and Sanparith Marukatat. Counterfactual mean embeddings. Journal of Machine Learning Research, 22:1–71, 2021.
>
> [6] Edward H. Kennedy. Towards optimal doubly robust estimation of heterogeneous causal effects. Electronic Journal of Statistics, 17:3008–3049, 2023.
>
> [7] Dennis Frauen, Fergus Imrie, Alicia Curth, Valentyn Melnychuk, Stefan Feuerriegel, Mihaela van der Schaar. A neural framework for generalized causal sensitivity analysis. In International Conference on Learning Representations, 2023.
>
> [8] Miruna Oprescu, Jacob Dorn, Marah Ghoummaid, Andrew Jesson, Nathan Kallus, and Uri Shalit. B-learner: Quasi-oracle bounds on heterogeneous causal effects under hidden confounding. In International Conference on Machine Learning, 2023
>
> [9] Duarte, G., Finkelstein, N., Knox, D., Mummolo, J., and Shpitser, I. An automated approach to causal inference in discrete settings. Journal of the American Statistical Association, 2023.
>
> [10] Bryan Lim, Ahmed M. Alaa, and Mihaela van der Schaar. Forecasting treatment responses over time using recurrent marginal structural networks. In NeurIPS, 2018.
>
> [11] Ioana Bica, Ahmed M. Alaa, James Jordon, and Mihaela van der Schaar. Estimating counterfactual treatment outcomes over time through adversarially balanced representations. In ICLR, 2020.
>
> [12] Dennis Frauen, Konstantin Hess, and Stefan Feuerriegel. Model-agnostic meta-learners for estimating heterogeneous treatment effects over time. arXiv preprint, 2024.
>
> [13] Fredrik Johansson, Uri Shalit, and David Sontag. Learning representations for counterfactual inference. In ICML, 2016.
>
> [14] Apperloo, E.M., Gorriz, J.L., Soler, M.J. *et al.* Semaglutide in patients with overweight or obesity and chronic kidney disease without diabetes: a randomized double-blind placebo-controlled clinical trial. Nature Medicine, 2024.
>
> [15] Stefanie Schüpke et al. Ticagrelor or Prasugrel in Patients with Acute Coronary Syndromes. The New England Journal of Medicine, 2019.
>
> [16] Ahmed Allam, Stefan Feuerriegel, Michael Rebhan, and Michael Krauthammer. Analyzing patient trajectories with artificial intelligence. Journal of Medical Internet Research, 23(12):e29812, 2021.
>
> [17] Stefan Feuerriegel et al. Causal machine learning for predicting treatment outcomes. Nature Medicine, 30:958–968, 2024.
>
> [18] Samuel L. Battalio et al. Sense2Stop: A micro-randomized trial using wearable sensors to optimize a just-in-time-adaptive stress management intervention for smoking relapse prevention. Contemporary Clinical Trials, 109:106534, 2021.
>
> [19] Ioana Bica, Ahmed M. Alaa, Craig Lambert, and Mihaela van der Schaar. From real-world patient data to individualized treatment effects using machine learning: Current and future methods to address underlying challenges. Clinical Pharmacology and Therapeutics, 109(1):87–100, 2021.
>
> [20] Yilmazcan Özyurt, Mathias Kraus, Tobias Hatt, and Stefan Feuerriegel. AttDMM: An attentive deep Markov model for risk scoring in intensive care units. In KDD. 2021.
>
> [21] Roderick J. Little, and Donald B. Rubin. Causal Effects in Clinical and Epidemiological Studies Via Potential Outcomes: Concepts and Analytical Approaches. Annual Review of Public Health, 2000.
>
> [22] Alistair E. W. Johnson et al. MIMIC-III, a freely accessible critical care database. Scientific Data, 3(1):160035, 2016.

---

### Author Response · Authors · 2024-11-21
**Response to all reviewers**

Thank you very much for the constructive evaluation of our paper and your helpful comments! We addressed all of them in the comments below. Furthermore, we uploaded an updated version of our paper and marked all significant changes to the original version in **blue color**.

Our **main improvements and clarifications** are the following:



* **Clarification of our contribution over existing methods:** We spell out the novelty of our work more clearly: our work proposes a **novel end-to-end training procedure for estimating CAPOs over time**. In other words, we do **not** propose a new architecture but **an entirely new training procedure in an end-to-end-architecture**. To this end, we revised our Sections 2, 3 and 4, where we reiterate that existing neural methods either **(i)** completely lack proper adjustments or **(ii)** have problematic adjustment strategies. Further, we clarify that our transformer-based instantiation is only **one** possible choice, and point to our **G-LSTM ablation study in our Supplement G.1**. Importantly, our G-LSTM outperforms all baselines (including more advanced architectures like the causal transformer from Melynchuk et al 2022), which shows that the source of gain is our new end-to-end training algorithm.
* **Related work:** \
We revised our related work **Section 2**:  \
**(i)** We added more explanations on why methods for estimating average potential outcomes have a completely different purpose and are **not** applicable in our setting. \
We further added an **extended related** work to our paper in a **new Supplement A**: **(ii)** Therein,  we discuss methods from survival analysis based on pseudo-outcomes and clarify why they are also **not** applicable to our setting.  \
**(iii)** Finally, we provide a detailed discussion on how Q-learning in the reinforcement literature and G-computation relate and highlight important differences.
* **Difference to existing methods:** We added a new **Section 4.4** where we discuss the differences to our method in greater detail. We explicitly point to the limitations of existing neural approaches and how our method improves upon them.
* **Clarification on low-variance claims:** We clarified our low-variance claims. In particular, we did the following:
**(i)** We **removed** our previous “low-variance” terminology throughout the paper to avoid ambiguities. \
**(ii)** We showed mathematically that IPW adjustments as employed by RMSNs construct pseudo-outcomes with **larger variance** than pseudo-outcomes constructed by G-computation in **Proposition 3**. \
**(iii)** We further explained that G-Net requires estimating **high-dimensional probability distributions** which may lead to large estimation variance (see our **revised Section 4.4 and Supplement F)**.

* **Additional experiments based on real-world data:** We added new experiments based on real-world data in a **new Supplement H**. We use the MIMIC-III dataset, which gives measurements from intensive care units aggregated at hourly levels and we predict the effect of vasopressors and mechanical ventilation on diastolic blood pressure. Our GT has state-of-the-art performance. Thereby, we show that our GT can easily handle difficulties in predicting patient outcomes in real-world datasets.  \
This supports our claim that our end-to-end training algorithm is highly effective.
* **Discussion about medical applicability:** We provide a discussion of how our GT is applicable in medical practice in a **new Supplement B**. Therein, we also discuss why our assumptions are standard in the literature, why they are weaker than for methods in the static setting, and why they are plausible  (see our also **revised Section 3**).
* **Coefficient of variation:** We report an additional metric for our main experiments, namely, the coefficient of variation (see our **new Supplement I). **The results show that** our GT provides the most stable estimates** of the CAPOs.

We highlight all changes in the **updated PDF** in **blue color**. We specifically incorporated all changes (marked with **Action**) into the updated version of our paper. Given these improvements, we are confident that our paper provides valuable contributions to the causal machine learning literature and is a good fit for ICLR 2025.

---

### Meta-Review · Area_Chair_6XpH · 2024-12-11

**Metareview:**

In this paper, the authors propose a framework to estimate the CAPO in temporal settings based on a G-transformer and pseudo-outcome regression. The paper is well written and sound, providing synthetic and semi-synthetic experiments. The authors provided a revised version of their paper with extensive changes.

However, Reviewer i9bM mentioned a lack of novelty compared to [1]. The authors have argued that the goal of that paper is limited to APO, but the reviewer suggested a misunderstanding of [1], which also performs CAPO. The authors did not engage further. Based on my cursory overview of [1], reviewer i9bM is correct, and I believe that comparing with [1] is a key step in demonstrating the efficiency of the proposed method. In addition, I would suggest extending the experiments to include real-world datasets.

[1] Shirakawa, Toru, et al. "Longitudinal Targeted Minimum Loss-based Estimation with Temporal-Difference Heterogeneous Transformer." International Conference on Machine Learning. PMLR, 2024.

**Additional Comments On Reviewer Discussion:**

The question of contribution compared to [1] was discussed, which resulted in reviewer i9bM increasing their confidence score.

---

### Decision · Program_Chairs · 2025-01-22

Reject